# Mirror Flow Matching with Heavy-Tailed Priors for Generative Modeling on Convex Domains

**Yunrui Guan**
Department of CMOR
Rice University
Houston, TX 77005
yg83@rice.edu

**Krishnakumar Balasubramanian**
Department of Statistics
University of California, Davis
Davis, CA 95616
kbala@ucdavis.edu

**Shiqian Ma**
Department of CMOR
Rice University
Houston, TX 77005
sqma@rice.edu

## Abstract

We study generative modeling on convex domains using flow matching and mirror maps, and identify two fundamental challenges. First, standard log-barrier mirror maps induce heavy-tailed dual distributions, leading to ill-posed dynamics. Second, coupling with Gaussian priors performs poorly when matching heavy-tailed targets. To address these issues, we propose Mirror Flow Matching based on a *regularized mirror map* that controls dual tail behavior and guarantees finite moments, together with coupling to a Student-$t$ prior that aligns with heavy-tailed targets and stabilizes training. We provide theoretical guarantees, including spatial Lipschitzness and temporal regularity of the velocity field, Wasserstein convergence rates for flow matching with Student-$t$ priors and primal-space guarantees for constrained generation, under $\varepsilon$-accurate learned velocity fields. Empirically, our method outperforms baselines in synthetic convex-domain simulations and achieves competitive sample quality on real-world constrained generative tasks.

## 1 Introduction

Flow matching (Lipman et al., 2023; Liu et al., 2023c; Albergo et al., 2023; Albergo & Vanden-Eijnden, 2023; Tong et al., 2024; Chen & Lipman, 2024) has emerged as a powerful framework for generative modeling, unifying score-based diffusion and optimal transport approaches under a single perspective. The central idea in flow matching is to construct a continuous-time deterministic flow that transports a simple prior distribution (e.g., Gaussian) to a complex target distribution, by learning its velocity field. Formally, given random variables $X_0 \sim \pi_0$ and $X_1 \sim \pi_1$, both supported on $\mathbb{R}^d$, we seek a time-dependent vector field $v : \mathbb{R}^d \times [0, 1] \to \mathbb{R}^d$ such that the solution of the ODE $dX_t = v(X_t, t)\, dt$, with $X_0 \sim \pi_0$, satisfies $X_1 \sim \pi_1$. A simple construction is based on straight-line interpolation $X_t = (1 - t)X_0 + tX_1$, which yields the conditional velocity field $v^*(x, t) = \mathbb{E}[X_1 - X_0 \mid X_t = x]$. This vector field $v^*$ minimizes the regression loss $\min_v \ \mathbb{E}\|v(X_t, t) - \frac{d}{dt}X_t\|^2]$, making it the optimal velocity field for the interpolation path. Since computing $v^*$ exactly is intractable, modern flow-matching methods approximate $v$ with a neural network and simulate the ODE numerically. This pathwise formulation leads to scalable training objectives, principled continuous-time generative processes, and improved sample quality.

**Constrained Flow Matching.** In many applications, the target is supported on constrained domains like polytope, simplex, or positive semidefinite matrices, rather than the full Euclidean space. Examples include molecular generation, where atoms and bonds must satisfy physical stability constraints (Fishman et al., 2023b), preference alignment (Kim et al., 2024), policy optimization and physical constraints for robotics (Zhang et al., 2025; Utkarsh et al., 2025) and watermarked content generation (Liu et al., 2023a). Standard flow-based methods fail in this setting: projecting unconstrained samples back onto the domain distorts the distribution.

**Related works.** Several strategies address the challenge in constrained flow matching, including reflection-based methods (Lou & Ermon, 2023; Fishman et al., 2023a; Xie et al., 2024; Christopher et al., 2024) that keep trajectories inside the domain using boundary normals; mirror-map diffusion

models (Liu et al., 2023a; Feng et al., 2025) that transform constrained problems into unconstrained ones using mirror-maps; gauge-map approaches (Li et al., 2025) that enforce feasibility via reflections; and distance-penalty methods (Huan et al., 2025; Khalafi et al., 2024) that penalize distance to the constraint set, at notable computational cost. Despite this progress, no framework yet ensures constraint satisfaction while providing convergence rates for flow matching.

In this work, we focus on the development of *mirror flow matching*, where the velocity field is adapted to the geometry of the constraint set. Formally, let $\mathcal{K} = \{\phi_i(x) < 0, \ \phi : \mathbb{R}^d \to \mathbb{R}, \ i = 1, \ldots m\}$, where $\phi_i$ are smooth convex functions, be a closed convex set, and suppose the target distribution $\pi_1$ is supported on $\mathcal{K}$. Our approach is based on constructing a mirror map $\nabla\Psi : \mathcal{K} \to \mathbb{R}^d$, where $\Psi : \mathcal{K} \to \mathbb{R}$ is a strictly convex, differentiable potential. The mirror map transports points from the constrained *primal* space $\mathcal{K}$ to an unconstrained *dual* space. In this dual space, one can perform standard (unconstrained) flow matching, i.e., define $Z_t = \nabla\Psi(X_t)$, and evolve it via $dZ_t = v^D(Z_t, t) \, dt$ with $Z_0 = \nabla\Psi(X_0)$, where $v^D$ is a velocity field learned by minimizing the unconstrained flow matching objective. The primal trajectory is then recovered by mapping back using the inverse mirror map $X_t = (\nabla\Psi)^{-1}(Z_t)$. This mirror-descent-based formulation ensures that the entire trajectory $\{X_t\}_{t \in [0,1]}$ remains in $\mathcal{K}$ while leveraging the flexibility of unconstrained flow matching in the dual space. Thus, mirror flow matching combines geometry-aware sampling with scalable learning, broadening the applicability of flow models to structured domains that naturally arise in the aforementioned application areas.

## 1.1 CHALLENGES AND SOLUTIONS

**Methodological Challenges.** Extending flow matching to constrained domains via mirror maps introduces key challenges. First, the transformed target distribution in the dual space may have heavy tails, causing standard mirror maps (e.g., log-barrier) to violate moment conditions required for well-posed flow ODEs (Figure 1, red dots). We address this with a *regularized mirror map* that controls heavy tails and ensures finite $p$-th moments for all $p \geq 1$ (Figure 1, blue dots), stabilizing training. Second, Gaussian priors often mismatch the heavy-tailed dual distributions; we instead adopt a *Student-$t$ prior*, improving alignment, sample quality, and stability. Together, these modifications overcome limitations of standard log-barrier and Gaussian priors, yielding high-fidelity constrained generative modeling. A visual illustration is provided in Appendix Section A.

**Theoretical Challenges.** In addition to the methodological issues above, theoretical analysis of mirror flow matching poses challenges. Rigorous error bounds for the sampling stage require the velocity field $v(x, t)$ to be Lipschitz in $x$ (Benton et al., 2024; Bansal et al., 2024; Zhou & Liu, 2025; Gao et al., 2024), while ODE discretization error further requires Lipschitz continuity in both $x$ and $t$ (Bansal et al., 2024; Zhou & Liu, 2025). However, the dual velocity field $v^D(z, t)$ is generally not Lipschitz over $t \in [0, 1]$. Partial progress includes spatial Lipschitzness on $t \in [0, T] \subsetneq [0, 1]$ under bounded $\pi_1$ (Benton et al., 2024; Zhou & Liu, 2025) or Gaussian-like $\pi_1$ (Gao et al., 2024). In general, unbounded $\pi_1$ can induce polynomial growth in $\|\nabla_x v(x, t)\|$ as $\|x\|$ grows and singularities near $t = 1$, motivating *early stopping*. Recent work (Cordero-Encinar et al., 2025) leverages Log-Sobolev inequalities to establish spatial Lipschitzness, though $t$-Lipschitzness is not addressed. We overcome this challenge by using t-distribution as priors. While such priors have been explored empirically (for example, (Pandey et al., 2025, Appendix B)), our motivation comes from addressing the above theoretical challenge.

**Contributions.** In this work, we introduce flow matching with a Student-$t$ prior (see Section 3) and provide new theoretical guarantees establishing both spatial Lipschitzness and temporal regularity (see Proposition 4.1). This result enables us to obtain explicit error bounds under substantially more general target distributions (see Theorem 3) in the dual Euclidean space under the assumption that the learned velocity fields approximates the true dynamics up to $\varepsilon$-accuracy. Finally in Theorem 4 we further prove *primal-space guarantees* for constrained dynamics.

## 2 INGREDIENTS FOR DESIGNING MIRROR FLOW MATCHING

### 2.1 INGREDIENT 1: THE MIRROR MAP

Before introducing our proposed mirror map, we first explain why the classical log-barrier is not suitable in our setting. The main issue arises from our first identified challenge: ensuring the existence

of moments. As the following general result shows, if the log-barrier transformation induces heavy tails, then even low-order moments (e.g., the second moment) may fail to exist.

**Lemma 2.1.** *Let $Y$ be a random variable in $\mathbb{R}^d$ with law $P$. Then, (i) if $P(\|Y\| \geq R) \geq C/R^p$ for some constant $C > 0$, then $\mathbb{E}[\|Y\|^p]$ does not exist, and (ii) if $P(\|Y\| \geq R) \leq C/R^\beta$ with $\beta > p$, then $\mathbb{E}[\|Y\|^p]$ is finite.*

In addition to controlling tails, we would also like the geometry induced by the mirror map to have a desirable metric property: the metric in the dual space should be *stronger* than that in the primal space. Formally, we require

$$\|x - y\| \leq L_\Psi \|\nabla\Psi(x) - \nabla\Psi(y)\|, \qquad \forall x, y \in \mathcal{K}, \tag{1}$$

for some constant $L_\Psi > 0$. To see why this is important, we first recall some definitions of $p$-Wasserstein distance in primal space and dual space. Let $\nu, \mu$ be two probability measures on $\mathcal{K}$. Then we have:

$$W_p(\nu, \mu)^p = \inf_{\gamma \in \Gamma(\nu,\mu)} \mathbb{E}_\gamma[\|x - y\|^p],$$

$$W_{p,\Psi}(\nu, \mu)^p = \inf_{\gamma \in \Gamma(\nu,\mu)} \mathbb{E}_\gamma[\|\nabla\Psi(x) - \nabla\Psi(y)\|^p],$$

where $\gamma \in \Gamma(\nu, \mu)$ means $\gamma$ is a coupling of $\nu, \mu$. The first one is just the Wasserstein distance for $\mathcal{K}$ under Euclidean distance, and the second one is actually the Wasserstein distance in the dual space. To see this, let $\nu', \mu'$ denote the distribution of $\nu, \mu$ in dual space, i.e., $\nu' = (\nabla\Psi)_\# \nu$ and $\mu' = (\nabla\Psi)_\# \mu$. Then we have $W_{2,\Psi}(\mu, \nu) = W_2(\mu', \nu')$. We remark that $W_{2,\Psi}$ was used to analyze the convergence of mirror Langevin algorithm (e.g., see Li et al. (2022)).

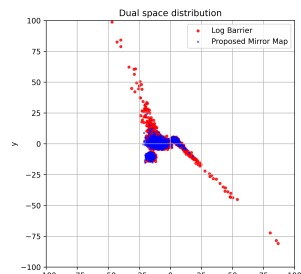

Figure 1: Dual space distribution comparison between the log barrier and our mirror map ($\kappa = 0.5$). The primal distribution is a truncated Gaussian mixture within a polytope (see Appendix A). The log barrier yields a heavy-tailed distribution, while our mirror map produces a much lighter tail.

In general, an upper bound for $W_{2,\Psi}(\nu, \mu)$ doesn't directly imply an error bound for $W_2(\nu, \mu)$ in the primal space. But under inequality (1), Wasserstein distances in the primal space can be controlled by those in the dual space:

$$W_2(\nu, \mu)^2 = \inf_{\gamma \in \Gamma(\nu,\mu)} \mathbb{E}_\gamma[\|x - y\|^2] \leq \inf_{\gamma \in \Gamma(\nu,\mu)} \mathbb{E}_\gamma[L_\Psi^2 \|\nabla\Psi(x) - \nabla\Psi(y)\|^2] = L_\Psi^2 W_{2,\Psi}(\nu, \mu)^2.$$

Inequality (1) is equivalent to $\nabla\Psi^*$ being $L_\Psi$-Lipschitz. Since $\nabla^2\Psi$ and $\nabla^2\Psi^*$ are inverses of each other, this condition is in turn equivalent to $\Psi$ being strongly convex. However, classical mirror maps are generally only *strictly* convex, not strongly convex. As a result, $L_\Psi$ can be arbitrarily large in certain domains; for instance, even for simple 2D polytopes with three facets ($d = 2, m = 3$), the constant $L_\Psi$ may blow up (see Example 5 in the Appendix).

These observations suggest that we need to design a new mirror map that balances tail behavior and convexity. In particular, the desired mirror map should satisfy the following goals:

1. Transform the constrained distribution into an unconstrained distribution on $\mathbb{R}^d$.
2. Ensure that key moments (e.g., the second moment) of the transformed distribution exist.
3. Be strongly convex, so that convergence guarantees in the dual Euclidean metric can be transferred to guarantees in the primal Euclidean metric.

Motivated by the mirror-map framework of Vural et al. (2022), we propose in Proposition 2.2 a *modified log-barrier* that achieves these properties.

**Proposition 2.2.** *Let $\mathcal{K} = \{\phi_i(x) < 0, \forall i \in [m]\}$, where $\phi_i$ are smooth convex functions with bounded gradient. Let $\Psi(x) = -\frac{1}{1-\kappa}\sum_{i=1}^m (-\phi_i(x))^{1-\kappa} + \frac{1}{2}\|x\|^2$. Then we have $W_2(\nu, \mu) \leq W_{2,\Psi}(\nu, \mu)$. Denote $\mathcal{K}_\delta = \{x \in \mathcal{K} : -\phi_i(x) \geq \delta\}$. Let $X$ be a random variable on $\mathcal{K}$ whose law is denoted as $P$. Assume there exists positive constants $C_\mathcal{K}, \beta, \delta_0$ s.t. for all $0 < \delta < \delta_0$ it holds that $P(\mathcal{K}\backslash\mathcal{K}_\delta) \leq C_\mathcal{K}\delta^\beta$. Then there exists some constant $C$ s.t. in the dual space $\mathbb{R}^d$, for all $R \geq C'/\delta_0^\kappa$ (here $C'$ is some constant that depends on $\mathcal{K}$), $P(\|\nabla\Psi(X)\| \geq R) \leq C/R^{\beta/\kappa}$. By choosing $\kappa < \beta/p$, we can guarantee $\mathbb{E}[\|\nabla\Psi(X)\|^p]$ exists.*

Specific examples (including $L_2$ ball and polytopes) are discussed in Appendix Section B. We verify that the boundary-measure condition $P(\mathcal{K} \setminus \mathcal{K}_\delta) \leq C_\mathcal{K} \delta^\beta$ is natural in typical cases.

**Example 1** (Uniform distribution on the cube). Let $\mathcal{K} = [-1, 1]^d$ and let $P$ be the uniform distribution on $\mathcal{K}$. Define the $\delta$-interior as $\mathcal{K}_\delta = \{x \in \mathcal{K} : d(x, \partial\mathcal{K}) \geq \delta\}$. Then the boundary layer has probability mass $P(\mathcal{K} \setminus \mathcal{K}_\delta) = \frac{2^d - (2-2\delta)^d}{2^d} = 1 - (1-\delta)^d$. Using the first-order expansion $(1-\delta)^d \approx 1 - d\delta$, we obtain $P(\mathcal{K} \setminus \mathcal{K}_\delta) \approx d\delta$. Hence the condition $P(\mathcal{K} \setminus \mathcal{K}_\delta) \leq C_\mathcal{K} \delta^\beta$ holds with $\beta = 1$ and $C_\mathcal{K} = d$. This shows the assumption is mild and satisfied by standard convex bodies such as the cube under uniform measure.

## 2.2 Ingredient 2: The Prior Distribution

For flow matching, let the target distribution be denoted by $X_1 \sim \pi_1$ with density $p$, and let the initial distribution (prior) be $X_0 \sim \pi_0$. The evolution between $\pi_0$ and $\pi_1$ is described by a time-dependent vector field, where $v(x, t)$ denotes the true vector field. Considering straight-line interpolation, by definition, the velocity field at a point $(x, t)$ is the conditional expectation of the instantaneous displacement along this interpolation: $v(x, t) = \mathbb{E}[X_1 - X_0 \mid X_t = x]$. To make this expression explicit (Karras et al., 2022; Wan et al., 2025), note that the interpolation relation $X_t = (1-t)X_0 + tX_1$ can be inverted to obtain $X_0 = \frac{1}{1-t}(X_t - tX_1)$. Substituting this into the displacement $X_1 - X_0$ yields $X_1 - X_0 = -\frac{1}{1-t}X_t + \frac{1}{1-t}X_1$. Taking conditional expectation given $X_t = x$, we obtain the closed-form expression for the true velocity field:

$$v(x, t) = \mathbb{E}\left[-\frac{1}{1-t}X_t + \frac{1}{1-t}X_1 \,\middle|\, X_t = x\right] = -\frac{1}{1-t}x + \frac{1}{1-t}\mathbb{E}[X_1 \mid X_t = x].$$

Thus, the vector field $v(x, t)$ consists of two interpretable terms: a deterministic contraction term $-\frac{1}{1-t}x$ that pulls $x$ toward the origin, and a prediction term $\frac{1}{1-t}\mathbb{E}[X_1 \mid X_t = x]$ that directs the flow toward the target distribution $\pi_1$.

A crucial modeling choice in flow matching is the prior distribution. The choice of the prior distribution affects this conditonal expectation $\mathbb{E}[X_1 \mid X_t = x]$ significantly. While Gaussian priors are the standard choice in unconstrained generative modeling, they are poorly suited when the target distribution exhibits heavy tails. The following example illustrates this pathology. Denote standard Student t distribution as $t_{d,\nu}(x) = C_{\nu,d}(1 + \frac{1}{\nu}\|x\|^2)^{-\frac{\nu+d}{2}}$.

**Example 2.** Consider the one-dimensional target density $X_1 \sim p(x) \propto (1 + \frac{1}{2}x^2)^{-\frac{3}{2}}$. Suppose we use a Gaussian prior $X_0 \sim \mathcal{N}(0, 1)$. Then the conditional distribution of $X_1$ given an interpolated point $X_t = x$, is given by

$$p(X_1|X_t = x) \propto g(x_1) := \exp\left(-\frac{(tx_1 - x)^2}{2(1-t)^2}\right)\left(\frac{1}{1 + \frac{1}{2}x_1^2}\right)^{\frac{3}{2}}.$$

This conditional distribution develops two modes: one near $x_1 = 0$ and another near $x_1 \approx x/t$. Although the $t \to 0$ limit will not cause a singularity (Wan et al., 2025), we emphasize that for large values of $\|x\|$, the vector field would scales as $\exp(x^2)$ for some small values of $t$, implying that the true velocity field $v(x, t)$ can blow up super-exponentially in $x$. Furthermore, as discussed in Wan et al. (2025); Zhou & Liu (2025), singularities exist as $t \to 1$. By contrast, if we replace the Gaussian prior with a heavy-tailed Student-$t$ prior (e.g., with $\nu = 1$), the conditional density becomes

$$p(X_1|X_t = x) \propto g(x_1) = \left(1 + \left\|\frac{x - tx_1}{1-t}\right\|^2\right)^{-1}\left(\frac{1}{1 + \frac{1}{2}x_1^2}\right)^{\frac{3}{2}},$$

for which the dominant mode remains near $x_1 = 0$ even as $x$ being large, over $t \in [0, T] \subsetneq [0, 1]$. In this case, the conditional expectation does not explode with $x$, and the resulting velocity field remains controlled. See Appendix Section C for a visualization.

This example highlights a key principle: when the target distribution is heavier-tailed than the prior, the conditional distribution is likely to have a mode that is dominant near $\frac{x}{t}$ for some values of $t$. Then the induced velocity field can diverge at large $\|x\|$, producing ill-posed dynamics and complicating error analysis. In particular, such blow-ups directly cause the Lipschitz constant of $v(x, t)$ to diverge

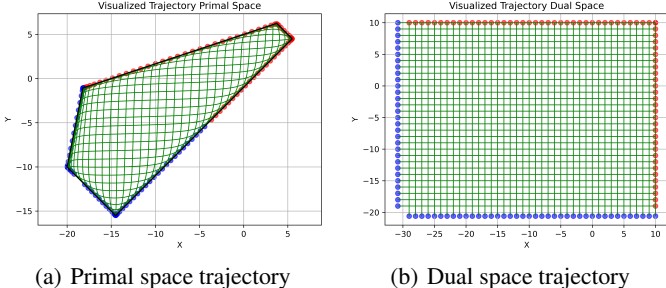

(a) Primal space trajectory     (b) Dual space trajectory

Figure 2: Visualization of interpolations in primal and dual spaces – Straight line interpolation in the dual space (Figure (b)) corresponds to curved "geodesic" interpolation in primal space Figure (a)).

as $\|x\| \to \infty$, necessitating additional assumptions on the tail of data distribution (e.g., bounded support) (Benton et al., 2024; Bansal et al., 2024; Gao et al., 2024; Zhou & Liu, 2025). Choosing a Student-$t$ prior prevents these blow-ups by making the data distribution to dominate the tail behavior of the conditional distribution, suppressing the mode near $x/t$. In this way, the mode near zero will be dominant, ensuring controlled velocity fields, finite-moment guarantees of the interpolation conditional distribution, and stability in both theoretical analysis and practical training.

## 3 MIRROR FLOW MATCHING

Recall from Section 2 that we discussed choices of mirror maps for closed convex sets of the form $\mathcal{K} = \{x \in \mathbb{R}^d : \phi_i(x) < 0, \ \forall i \in [m]\}$. In mirror flow matching, both the prior $\pi_0$ and the target $\pi_1$ are required to be supported on $\mathcal{K}$. The objective is to learn a continuous-time flow $X_t$ defined by the ODE $\frac{d}{dt}X_t = v^P(X_t, t)$ with $X_0 \sim \pi_0(x)$ that transports $\pi_0$ to $\pi_1$ over the interval $t \in [0, 1]$.

Mirror flow matching achieves this transport by interpolating in a transformed (mirror) space. Given a mirror map $\nabla\Psi$, we map $x \in \mathcal{K}$ into the dual space via $z = \nabla\Psi(x)$. As shown in Li et al. (2022), the dual Euclidean space $(\mathbb{R}^d, I_d)$ is isometric to the primal space equipped with the squared Hessian metric $(\mathcal{K}, (\nabla^2\Psi)^2)$. We denote these metrics as $g^P = (\nabla^2\Psi)^2$ and $g^D = I_d$. The procedure is then as follows: (1.) Map primal data to dual space: $z = \nabla\Psi(x)$. (2.) Perform flow matching in the dual space using straight-line interpolation $Z_t = (1-t)Z_0 + tZ_1$. (3.) After generating samples $\hat{z}$ in the dual space, map them back to primal space using the inverse mirror map $\hat{x} = \nabla\Psi^*(\hat{z})$. In particular, interpolation in primal space is defined as $X_t = \nabla\Psi^*(Z_t)$, which can be interpreted as the *geodesic interpolation* between $X_0$ and $X_1$ under the squared Hessian metric. See Figure 2 for an illustrative trajectory visualization in both the primal and dual spaces.

**Relation between dual and primal velocity fields.** Consider a dual-space flow $Z_t$ defined by vector field $v^D$. By direct differentiation, the corresponding primal velocity field is

$$v^P(X_t, t) := \frac{d}{dt}X_t = \nabla^2\Psi^*(Z_t)\left(\frac{d}{dt}Z_t\right) = \nabla^2\Psi^*(Z_t)v^D(Z_t, t). \tag{2}$$

The flow matching objective in the dual space is

$$\min_v \mathbb{E}_{t, Z_0, Z_1}\left[\|v^D(Z_t, t) - \tfrac{d}{dt}Z_t\|^2_{g^D}\right], \qquad Z_t = (1-t)Z_0 + tZ_1, \tag{3}$$

whose solution is known to be the conditional expectation $v^D(z, t) = \mathbb{E}[\frac{d}{dt}Z_t \mid Z_t = z]$ (Liu et al., 2023b). The following proposition establishes the equivalence between primal and dual formulations.

**Proposition 3.1.** *Learning a vector field in the dual Euclidean space $(\mathbb{R}^d, I_d)$ is equivalent to learning a vector field in the primal space $(\mathcal{K}, (\nabla^2\Psi)^2)$. Specifically,*

$$\min_v \mathbb{E}\left[\|v^P(X_t, t) - \tfrac{d}{dt}X_t\|^2_{g^P}\right] \quad \text{and} \quad \min_v \mathbb{E}\left[\|v^D(Z_t, t) - \tfrac{d}{dt}Z_t\|^2_{g^D}\right]$$

*are equivalent, with the correspondence $v^D(z, t) = \nabla^2\Psi(x)v^P(x, t)$. Moreover, the primal flow matching objective is solved by $v^P(x, t) = \mathbb{E}\left[\frac{d}{dt}X_t \mid X_t = x\right]$.*

---

**Algorithm 1** Mirror Flow matching with Student t distribution

---

1: Map data distribution from $\mathcal{K}$ to $\mathbb{R}^d$ using $\nabla\Psi$, obtain samples for $Z_1$.
2: Learn a vector field $\hat{v}^D(z,t)$ with prior $\pi_0(x) \sim t_{d,\nu}$ via $\min_{\hat{v}^D} \mathbb{E}_{t,Z_0 \sim \pi_0^D, Z_1 \sim \pi_1^D} \left[ \|\hat{v}^D(Z_t,t) - (Z_1 - Z_0)\|^2 \right]$ where $Z_t = tZ_1 + (1-t)Z_0$.
3: Choose step size $h$ for Euler discretization s.t. $\frac{1}{h}$ is integer. Choose $T \in (0,1)$ as early stopping time, satisfying $\frac{T}{h} \in \mathbb{Z}$.
4: Perform Euler discretization to sample from $\pi_1^D$ with constant step size $h$, up to time $T$:
5: Generate $\overline{z}_0 \sim \pi_0^D$.
6: **for** $k = 0$ to $\frac{T}{h} - 1$ **do**
7: $\quad \overline{z}_{h(k+1)} = \overline{z}_{hk} + h\hat{v}(\overline{z}_k, hk)$
8: **end for**
9: Denote the obtained sample by $\overline{z}_T \sim \hat{\pi}_T^D$.
10: Map samples $\overline{z}_T$ back to $\mathcal{K}$ using $\nabla\Psi^*$ to obtain $\overline{x}_T$.

---

This result shows that training in the dual space with straight-line interpolation is equivalent to training in the primal space with geodesic interpolation under the squared Hessian metric. From an algorithmic standpoint, this equivalence is highly convenient: we can train the dual-space vector field $v^D$, which is simpler due to its Euclidean geometry, and recover the primal vector field $v^P$ by the transformation in equation 2. Thus, the difficult geometry of $\mathcal{K}$ is automatically handled by the mirror map, while optimization is carried out in an unconstrained Euclidean space.

The algorithmic procedure for mirror flow matching is summarized in Algorithm 1. This pipeline leverages the simplicity of Euclidean training in dual space, while ensuring that the generated samples respect the original convex constraints in primal space. Here, $h$ denotes the step size (in sampling stage) and $T < 1$ denotes the terminal time if early stopping is adopted.

## 4 THEORETICAL RESULTS

In this section, we provide a theoretical analysis of error bounds for flow matching. A key component of our analysis is the accuracy of the neural network used to approximate the target velocity field. We adopt the following assumption, which is standard in the literature on flow-based generative modeling (see, e.g., Benton et al. (2024); Bansal et al. (2024); Li et al. (2025)) as well as in the study of diffusion models (see, e.g., Chen et al. (2023); Li et al. (2024)). Theoretical justification for this assumption can be found in Wang et al. (2024); Zhou & Liu (2025), where the authors establish that such an $\varepsilon$-level approximation error can be achieved by a neural network under suitable training conditions.

**Assumption 1.** *(Neural Network Estimation Error) Let $v(x,t)$ denote the true velocity field and $\hat{v}(x,t)$ its neural network approximation. We assume that the approximation error is bounded in mean square, i.e., $\mathbb{E}\left[\|v(x,t) - \hat{v}(x,t)\|^2\right] \leq \varepsilon^2$.*

Intuitively, Assumption 1 states that the learned velocity field $\hat{v}$ is close to the true velocity field $v$ in an average sense across both space and time. The parameter $\varepsilon$ therefore quantifies the quality of the neural network approximation: smaller $\varepsilon$ implies a more accurate approximation, which directly translates into higher fidelity of the generated samples.

### 4.1 GUARANTEES FOR EUCLIDEAN FLOW MATCHING WITH t-DISTRIBUTION PRIORS

In this subsection, we provide an error analysis for flow matching in Euclidean space when the prior distribution is chosen to be a Student-$t$ distribution (henceforth referred to as *t-Flow*). Our analysis applies to the general framework of flow matching with straight-line interpolation, and is not restricted to the mirror flow matching setup. To maintain notation consistency, we denote random variables as $Z \in \mathbb{R}^d$ with density $\pi_1^D$. We begin by introducing the assumptions required.

**Assumption 2** (Finite Moments). *Let $Z_0$ denote the prior (chosen as Student-t) random variable and $Z_1$ denote the target random variable, both supported on $\mathbb{R}^d$. We assume that they have finite second moments, i.e., $\mathbb{E}[\|Z_0\|^2] < \infty, \mathbb{E}[\|Z_1\|^2] < \infty$, which is necessary for well-definedness.*

**Assumption 3** (Polynomial Tail Bound). *Let $\pi_1^D(x)$ denote the probability density function of the data distribution supported on $\mathbb{R}^d$. It is assumed to satisfy: (1) For $\|x\| \geq 1$, we have $\pi_1^D(x) \leq \frac{C}{\|x\|^\alpha}$, and (2) For $\|x\| < 1$, we have $\pi_1^D(x) \leq C_u$.*

The above assumption allows the target distribution to be heavy-tailed, covering a wide range of realistic distributions. We next establish Lipschitz guarantees for the true vector field, showing that under Assumption 3, the velocity field induced by t-Flow is both spatially Lipschitz and admits a controlled temporal derivative, which is crucial for bounding the discretization error in Theorem 3.

**Proposition 4.1.** *Let $v^D$ be the minimizer of the t-Flow objective (Equation 3). Under Assumption 3 with $\alpha \geq 2d + \nu + 2$, there exist constants $B_1, B_2$ such that, for all $t \in [0, T]$:*
*1. (Spatial Lipschitzness) The vector field $v^D(z, t)$ is $L_1$-Lipschitz in $z$, with $L_1 := \frac{d+\nu}{(1-T)^2} B_1$.*
*2. (Temporal Regularity) The time derivative of the velocity field is bounded as*

$$\left\| \frac{\partial}{\partial t} v(z, t) \right\| \leq \frac{1}{(1-T)^2} \|z\| + \frac{1}{(1-T)^2} B_1 + \frac{1}{1-T} \frac{\nu+d}{\nu} \frac{3\sqrt{\nu}}{2(1-T)^2} \left( B_2 + 3B_1^2 \right).$$

The proof is deferred to Appendix G.1. To the best of our knowledge, the only prior work that controlled the temporal Lipschitzness of the vector field in order to bound discretization error is Zhou & Liu (2025). However, their analysis required the data distribution to have bounded support, whereas our result only assumes a polynomial tail bound. For spatial Lipschitzness, existing results either imposed stronger conditions on the data distribution (Zhou & Liu, 2025; Benton et al., 2024; Gao et al., 2024) or studied different problem settings (Cordero-Encinar et al., 2025). We can now quantify the discretization error of t-Flow.

**Theorem 3** (Discretization Error of t-Flow). *Consider t-Flow in Euclidean space. Let $\pi_1^D$ denote the data distribution supported on $\mathbb{R}^d$, and $\hat{\pi}_T^D$ be the law of generated sample $\overline{z}_T$ obtained by Euler discretization with constant step size $h$, up to time $T$ (see line 9 of Algorithm 1). Under Assumption 3 with $\alpha \geq 2d + \nu + 2$, Assumption 2, and Assumption 1, there exists a constant $D_3$, depending polynomially on $\frac{1}{1-T}$, $d$, $\nu$, and on $B_1, B_2, \mathbb{E}[\|Z_1\|^2], \mathbb{E}[\|Z_0\|^2]$, such that*

$$W_2(\pi_1^D, \hat{\pi}_T^D) \leq \frac{e^{6L_1}}{L_1} \sqrt{h^2 D_3 + \varepsilon^2} + (1-T)\sqrt{2\left(\mathbb{E}[\|Z_1\|^2] + \mathbb{E}[\|Z_0\|^2]\right)}.$$

The proof is provided in Appendix G.2. The error bound consists of two terms. The first term captures the discretization error (from Euler steps of size $h$) and the neural network approximation error (measured by $\varepsilon$). Both vanish as $h \to 0$ and $\varepsilon \to 0$. The second term corresponds to early stopping error, which decreases to zero as $T \to 1$. Thus, by taking $T$ close to 1 and ensuring accurate vector field approximation with sufficiently small step size, we can guarantee high-quality samples.

We now compare our result with recent related works. Bansal et al. (2024) did not analyze the Lipschitz properties of the velocity field, but instead imposed them as assumptions. Zhou & Liu (2025) established both spatial and temporal Lipschitzness and further analyzed neural network approximation, but required the data distribution to have bounded support. We note that the exponential dependence on the spatial Lipschitz constant $L_1$ arises due to non-convexity, and also appears in existing analyses (Bansal et al., 2024; Zhou & Liu, 2025). It is plausible that this exponential dependency could be improved to polynomial dependence by following the probabilistic coupling strategy in Chen et al. (2023), though the resulting algorithm is not purely deterministic.

### 4.2 Primal Space Guarantee for Mirror Flow Matching

We next obtain the following primal space guarantee. First note that that the primal space $(\mathcal{K}, g^P)$ and the dual space $(\mathbb{R}^d, g^D)$ are isometric. Hence, we have the following result.

**Lemma 4.2.** *If the vector field $v^D$ is $L_1$ Lipschitz in the dual space $(\mathbb{R}^d, g^D)$, it is $L_1$ Lipschitz in the primal space $(\mathcal{K}, g^P)$ (under the squared Hessian metric).*

To relate Assumption 3 with the distribution in primal space, we impose the following condition.

**Assumption 4.** *(Primal Space Probability Density Function). Denote $\pi_{Euc}^P(x)$ as the probability density function for $\pi_1^P$ in the primal space, under Euclidean metric. Assume that $\pi_{Euc}^P(x)$ is smooth and that there exists a small constant $\delta_0$ such that $\sup_{x \in \mathcal{K} \setminus \mathcal{K}_\delta} \pi_{Euc}^P(x) \leq C_{pdf} \delta^\gamma, \forall \delta \leq \delta_0$.*

**Theorem 4.** *Let $\hat{\pi}_T^P$ be the law of output samples generated by Algorithm 1 (i.e., the law of $\overline{x}_T$ in Line 10). Under Assumption 1 and 4, with $\kappa \leq \frac{\gamma}{2d+\nu+2}$, and we further require $\kappa < \frac{\beta}{2}$, there exists constant $L_1, D_3$ and $M := \sqrt{2\left(\mathbb{E}[\|Z_1\|^2] + \mathbb{E}[\|Z_0\|^2]\right)}$ such that*

$$W_2(\pi_1^P, \hat{\pi}_T^P) \leq \frac{e^{6L_1}}{L_1}\sqrt{h^2 D_3 + \varepsilon^2} + (1-T)M.$$

The proof is provided in Appendix G.3 and essentially follows by Proposition 2.2 and Theorem 3.

## 5 EXPERIMENTS

We demonstrate the effectiveness of our approach by performing numerical simulation (see section 5.1) and real world data experiments on AFHQv2 dataset (see section 5.2). The numerical simulation is performed on a personal laptop using a CPU. The real world data experiments were performed on a single A100 GPU.

### 5.1 NUMERICAL SIMULATION

We build on the experimental setup of Li et al. (2025) and conduct numerical simulations on two representative constrained generative modeling tasks. The first task is a 10-dimensional polytope problem, defined as $\{x \in \mathbb{R}^{10} : a_i^\top x < b_i, \ i = 1, 2, \ldots, 30\}$, with constraints loaded from a pre-specified data file from Li et al. (2025). The target distribution is a uniform mixture of Gaussians, where the means are partly sampled at random and partly human-designed to stress-test the model (e.g., $(-3, -3, 3, 3, \ldots, -3) \in \mathbb{R}^{10}$), and covariances are fixed to $0.4I_{10}$. The second task is a 6-dimensional $L_2$ ball problem, defined as $\{x \in \mathbb{R}^6 : \|x\|^2 < 25\}$, with target distributions generated in the same manner as in the polytope case. For both tasks, we used a simple MLP network with 4 layers, and hidden layer size being 128. We used ELU activation function. We perform 10,000 training iterations.

We implemented our method with $\kappa = 0.3$ and used a $t$-Flow prior with $\nu = 10$. As shown in Table 1 and Table 2, our approach consistently outperforms both Gauge Flow Matching (Li et al., 2025) and Reflected Flow Matching (RFM) (Xie et al., 2024). Across both tasks, our method achieves lower KL divergence and smaller Maximum Mean Discrepancy (MMD) values, while simultaneously guaranteeing sample feasibility. For the $L_2$ ball case, Gauge Flow Matching is omitted since it coincides with Reflected Flow Matching.

We also evaluated the performance of our approach under different choices of $\mu, \kappa$ for a 10 dimensional polytope task. The results are presented in Figure 3. In addition, we visualized the dual space distribution to justify that our mirror map doesn't induce heavy tail, whereas the log-barrier does; See Appendix D. Empirically, we observed that t-Flow outperforms G-Flow. Also, larger values of $\kappa$ would induce a tail that is heavier than smaller values of $\kappa$. The result indicates that a large $\nu$ would require a smaller $\kappa$, which is consistent with our theoretical findings.

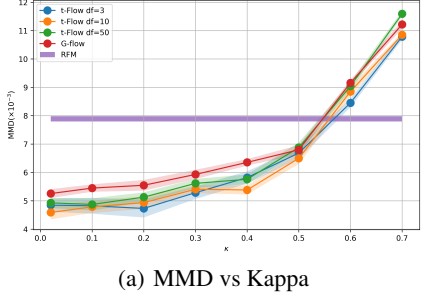

(a) MMD vs Kappa

Figure 3: We test the performance of t-Flow and G-Flow for different values of $\kappa$. For t-Flow, we compare the performance among different choices of $\nu$ (degree of freedom). We use RFM as baseline comparison. Networks are trained with 40,000 iterations.

Table 1: Performance comparison with 10-dimensional polytope constraints. Results are based on an average of 10 runs, each run with an average of $10,000$ samples. MMD values are scaled by $10^{-2}$.

| Method | MMD ↓ | KL Divergence ↓ | Feasibility |
|---|---|---|---|
| Mirror t-Flow | **$0.995 \pm 0.021$** | **$1.424 \pm 0.037$** | $100\%$ |
| Mirror G-Flow | $1.006 \pm 0.016$ | $1.447 \pm 0.046$ | $100\%$ |
| Gauge Vanilla (Li et al., 2025) | $1.828 \pm 0.011$ | $5.023 \pm 0.073$ | $95.257 \pm 0.150\%$ |
| Gauge Reflect (Li et al., 2025) | $1.830 \pm 0.011$ | $5.057 \pm 0.075$ | $100\%$ |
| RFM (Xie et al., 2024) | $1.217 \pm 0.007$ | $2.034 \pm 0.052$ | $100\%$ |
| MDM (Liu et al., 2023a) | $1.258 \pm 0.013$ | $1.708 \pm 0.054$ | $100\%$ |

Table 2: Performance comparison on 6-dimensional $L_2$ ball constraints. Results are based on an average of 10 runs, each run with an average of $10,000$ samples. MMD values are scaled by $10^{-2}$.

| Method | MMD ↓ | KL Divergence ↓ | Feasibility |
|---|---|---|---|
| Mirror t-Flow | **$5.329 \pm 0.101$** | **$0.162 \pm 0.011$** | $100\%$ |
| Mirror G-Flow | $6.244 \pm 0.286$ | $0.176 \pm 0.015$ | $100\%$ |
| RFM (Xie et al., 2024) | $5.935 \pm 0.222$ | $0.285 \pm 0.012$ | $100\%$ |
| MDM (Liu et al., 2023a) | $36.156 \pm 0.102$ | $8.017 \pm 0.046$ | $100\%$ |

We also implemented MDM (Liu et al., 2023a) for both tasks. We remark that in the original MDM paper, the author only provided a closed-form formula of their mirror map (and its inverse) under specific assumptions of the polytope, which can't be applied for an arbitrary polytope. For an implementation of log-barrier, the inverse mirror map is difficult to solve, and therefore we implemented MDM with regularized log-barrier. For $L_2$ ball case, we implemented MDM with the closed form mirror map provided in Liu et al. (2023a). We observed that the neural network failed to learn useful information, which is likely due to the heavy tailed nature of the inducede dual space distribution.

These experiments highlight the advantages of our method. By jointly choosing mirror maps and priors based on careful analysis, our approach achieves superior performance on numerical benchmarks while preserving feasibility by construction. The ability to obtain tighter divergence metrics under strict feasibility underscores its promise for high-dimensional constrained generative modeling, demonstrating robustness across geometries (polytope vs. $L_2$ ball) and scalability to practical domains where constraints are central.

## 5.2 REAL-DATA APPLICATION: WATERMARKED IMAGE GENERATION

Following Liu et al. (2023a), we evaluate our method on the task of $64 \times 64$ watermarked image generation using the AFHQv2 dataset. We begin by generating parameters $(a_i, b_i, c_i)$, which serve as user-specific private keys. These parameters define a polytope $\mathcal{K} = \{x : c_i < a_i^\top x < b_i\}$, where an image can be vectorized and checked for feasibility: an image lying inside $\mathcal{K}$ is verifiably generated by the model. During training, we first watermark the AFHQv2 images by projecting them (with added noise) onto the polytope, thereby producing a watermarked dataset. We then use these watermarked images as training data and compare the performance of Mirror Diffusion Models (MDM) (Liu et al., 2023a) with our proposed Mirror $t$-Flow approach.

A crucial component is the initialization used for the models. We first train both methods with random neural network initialization under a limited training budget (24 hours). We set the mirror map parameter as $\kappa = 0.1$ for our method, with random initialization. We first report the CMMD metric (Jayasumana et al., 2024). CMMD combines CLIP embedding with Maximum Mean Discrepancy metric and is considered more reliable than FID for evaluating generative models. With $10,000$ generated images, our approach achieves a CMMD score of $0.177$, which is competitive with the MDM baseline (Liu et al., 2023a), calculated to be $0.152$. Nevertheless, as shown in Figure 4(a), our method

Table 3: Performance comparison on watermarked image generation on the AFHQ2 dataset. Both implementations are initialized at EDM (Karras et al., 2022) checkpoint. For MDM, we use the code from Liu et al. (2023a). For flow matching, we apply the training framework from Lee et al. (2024).

| Method | FID (50k)↓ | CMMD | training time |
|---|---|---|---|
| Mirror Flow ($\kappa = 0.05$) | 4.27 | 0.023 | 3 hours |
| Mirrod Diffusion Model (Liu et al., 2023a) | 7.29 | 0.170 | 13 hours |

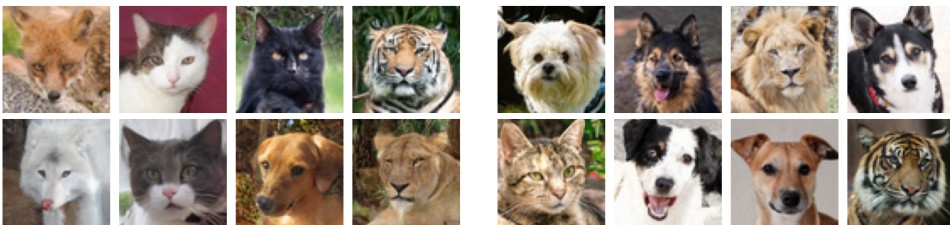

(a) With random initialization          (b) With EDM checkpoint initialization

Figure 4: Samples of generated watermarked images from the AFHQv2 $64 \times 64$ dataset. Constraint satisfaction were checked with built-in functions of Liu et al. (2023a).

already produces visually high-quality samples within a limited training budget, demonstrating strong potential for further improvements with better initializations.

Towards that, in Table 3 we next report results when the models are initialized at EDM (Karras et al., 2022) checkpoint for AFHQv2 dataset; the corresponding sample images are displayed in in Figure 4(b). We note that in this case, our method achieves superior CMMD and FID scores, requiring a smaller amount of training time. Finally, we remark that if we initialize at the checkpoint for a flow matching model from Lee et al. (2024), the FID (50k) can achieve $3.14$ after $1.5$ hours of training. This value is similar to $3.05$ reported in Liu et al. (2023a), while fully executing their scheduled number of iterations could result in an estimated training time up to several hundred hours in our experimental setup.

## 6 CONCLUSION

We introduced *t-Flow*, a flow-matching framework with Student-$t$ priors, and established rigorous guarantees on both spatial Lipschitzness and temporal regularity of the underlying velocity field . Our analysis yielded the first error bounds for flow matching under polynomial tail assumptions, thereby extending prior results beyond bounded-support assumptions. We further demonstrated that $t$-Flow provides robust sample quality in practice, particularly in scenarios where Gaussian priors fail to capture heavy-tailed structures. Beyond technical guarantees, our results emphasize that successful generative modeling on complex domains requires a careful co-design of mirror maps and priors, rather than defaulting to standard choices. This perspective opens up several promising avenues. One direction is exploring adaptive choices of degrees of freedom in the $t$-prior could yield even more flexibility, enabling flows that automatically adapt to local tail behavior of the data. Another is extending $t$-Flow to constrained domains with non-convex geometry, potentially leveraging landing techniques. On the theory front, improving the exponential dependence on Lipschitz constants, for example via probabilistic couplings or randomized flow strategies is interesting. Finally integrating $t$-Flow with hybrid diffusion–flow architectures and energy-based models offers yet another exciting path, combining the complementary strengths of these paradigms.

### ACKNOWLEDGMENTS

Research of Krishnakumar Balasubramanian was supported in part by National Science Foundation (NSF) grant DMS-2413426. Research of Shiqian Ma was supported in part by NSF grants CCF-2311275 and ECCS-2326591 and ONR grant N00014-24-1-2705.

## REPRODUCIBILITY STATEMENT

Proofs for the theoretical results are presented in the Appendix. Codes to reproduce the experimental results are provided in the supplementary material. **LLM usage:** LLM was used only to polish the writing.

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

## A  Visual Illustration of Methodological Challenges

We illustrate the benefits of our approach in Figure 5. The constraint set is a polytope $\mathcal{K} = \{x \in \mathbb{R}^2 : Ax < b\}$ with

$$A^\top = \begin{pmatrix} 1 & -1 & 1 & -5 & -1/3 \\ 1 & -1 & -1 & 1 & 1 \end{pmatrix}, \qquad b^\top = \begin{pmatrix} 10 & 30 & 1 & 90 & 5 \end{pmatrix}.$$

The target $\pi_1$ is a mixture of three Gaussians, truncated to $\mathcal{K}$: $\mathcal{N}([-10,0]^T, \mathrm{diag}\,(8,2))$ with weight 0.6, $\mathcal{N}([-15,-10]^T, \mathrm{diag}\,(1,1))$ with weight 0.2, and $\mathcal{N}([3,3]^T, \mathrm{diag}\,(0.5,0.25))$ with weight 0.2. We compare G-flow (Gaussian prior) and t-flow (Student-$t$ prior) under both the log-barrier mirror map and our proposed regularized map (Figures 5(b)–5(e)), alongside samples from the true target (Figure 5(a)). Vector fields were parameterized by neural networks and simulated via Euler discretization ($h = 0.1$) with early stopping. As shown in Figure 5, our approach achieves robust mode recovery and faithful constrained sampling, consistently outperforming Gaussian-based flow methods.

## B  Examples verifying Proposition 2.2

Proposition 2.2 can be specialized to several classical examples of convex sets.

1. $L_2$ **ball.** Consider the closed Euclidean ball $\mathcal{K} = \{x \in \mathbb{R}^d : \|x\| < R\}$. Define the mirror potential $\Psi(x) = -\frac{1}{1-\kappa}\left(R^2 - \|x\|^2\right)^{1-\kappa} + \frac{1}{2}\|x\|^2$. In this case the barrier function is $\phi(x) = \|x\|^2 - R^2$, which is clearly smooth and convex. Moreover, its gradient is bounded on $\mathcal{K}$, satisfying the required assumptions.

2. **Polytope.** Let $\mathcal{K} = \{x \in \mathbb{R}^d : a_i^T x < b_i, \forall i \in [m]\}$ be a polytope defined by $m$ linear inequalities. Define the potential $\Psi(x) = -\sum_{i=1}^m \frac{1}{1-\kappa}\left(b_i - a_i^T x\right)^{1-\kappa} + \frac{1}{2}\sum_{j=1}^d x_j^2$. Here the barrier functions are $\phi_i(x) = a_i^T x - b_i$. Each $\phi_i$ is affine (hence smooth and convex), with Hessian $\nabla^2 \phi_i(x) = 0$, and its gradient is bounded uniformly. Thus the conditions are again satisfied.

## C  Visual Illustration corresponding Section 2.2

We illustrate the blow-up phenomenon discussed in Section 2.2. In Figure 7(a), 7(b), 7(c) we illustrate the $t \to 0$ limit, blows-up for small $t$, and $t \to 1$ limit respectively, for the G-flow. The corresponding Figure 7(d), 7(e), 7(f) for the t-flow is more benign.

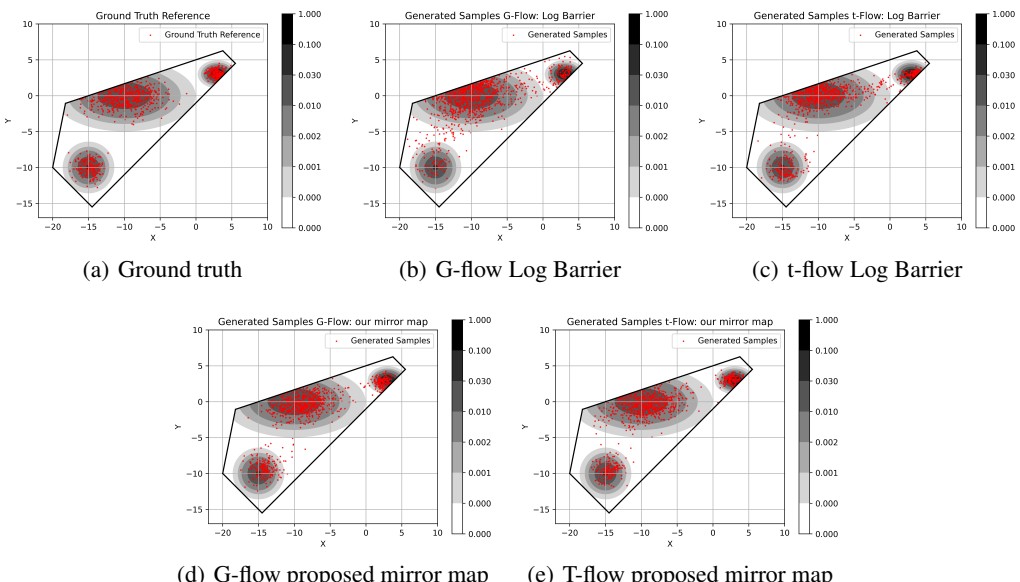

Figure 5: Figure 5(a) shows the ground-truth reference distribution. Figures 5(b) and 5(c) illustrate that the log-barrier method performs poorly (both with G or t-flow), while Figure 5(d) demonstrates that G-flow (with our mirror map) fails to capture the mode centered near $(-10, 0)$. In contrast, Figure 5(e) shows that t-flow with our mirror map covers the target distribution better. All results are obtained with discretization step size $h = 0.1$. See also Figure 6 for a zoomed-in illustration near the boundary.

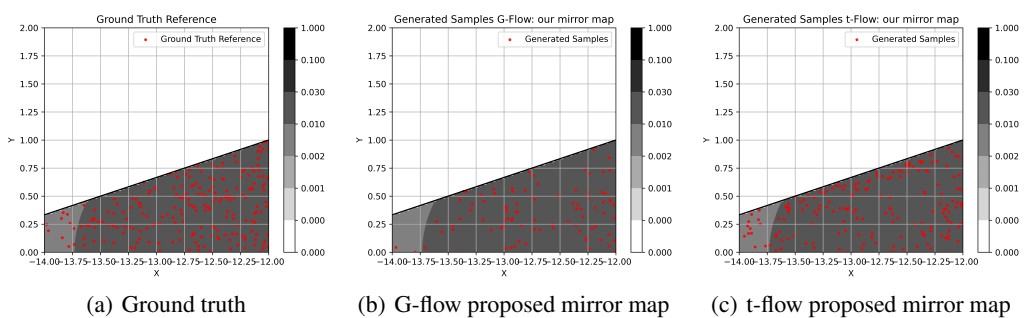

Figure 6: We generate a total of 10,000 samples, but for visualization we only display those lying in the boundary region $x \in [-14, -12], y \in [0, 2]$. Figure 6(a) shows the ground-truth reference distribution. Figures 6(b) and 6(c) demonstrate that, near the boundary, t-flow provides a closer approximation to the ground truth than G-flow.

# D ADDITIONAL EXPERIMENTAL RESULTS

We visualize the dual space distribution induced by different mirror maps. To plot the figure, we generate $10,000$ true samples inside a 10 dimensional polytope, and map them to a dual space using different mirror maps. Then we select a two-dimensional subspace $y_2 \times y_5$ of the dual space $\mathbb{R}^{10}$, and visualize the samples in this subspace. See Figure 8. We observe that the log-barrier indeed produces heavy tails, whereas our mirror map with $\kappa = 0.2$ doesn't.

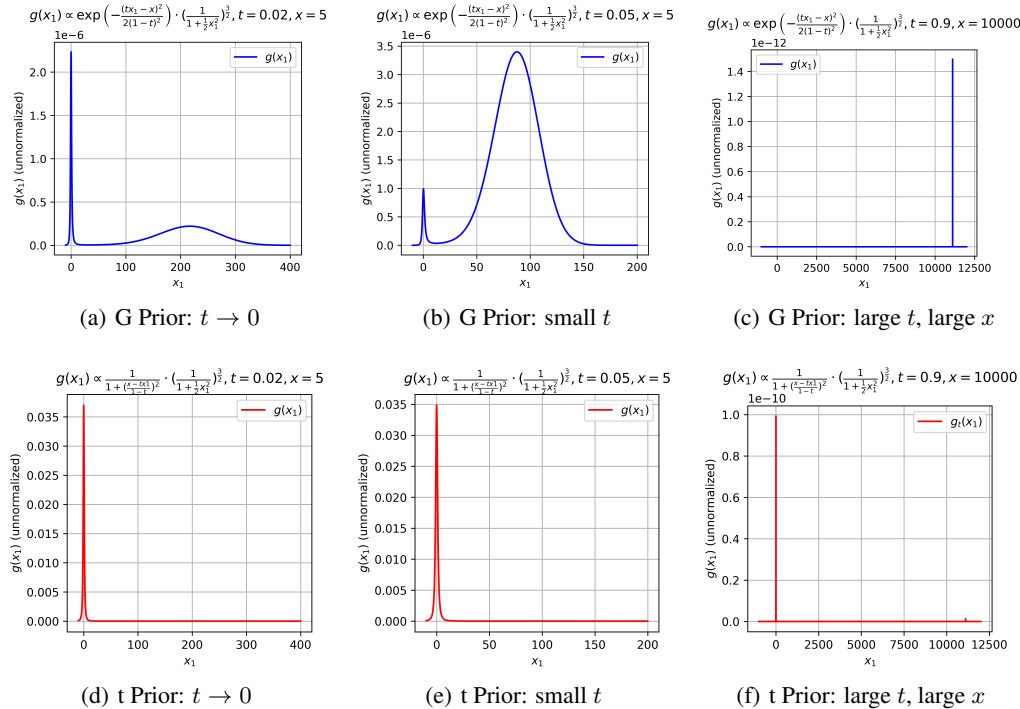

Figure 7: Illustration for Example 2. (i) Figures 7(a) and 7(d) demonstrate that in the limit $t \to 0$, the distribution remains well-behaved and does not blow up. (ii) Figure 7(b) shows that for sufficiently large values of $x$ (here we choose a moderately large $x$ for readability), there exists a small value of $t$ such that the flow with a Gaussian prior diverges. (iii) Figure 7(e) illustrates that such divergence does not occur when using a Student-$t$ prior. (iv) Finally, Figures 7(c) and 7(f) show that as $t$ approaches 1, the Gaussian-prior flow becomes unstable, whereas the Student-$t$ prior remains stable.

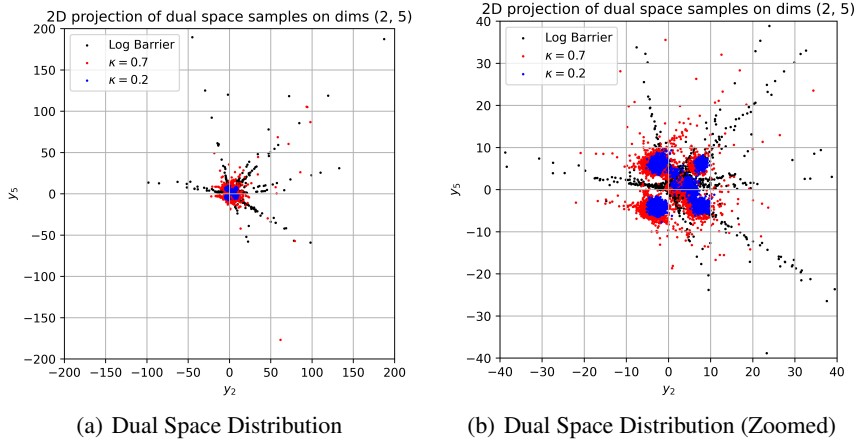

Figure 8: We compare the difference between our mirror map (with different $\kappa$), and the log-barrier.

## E    PROOFS FOR SECTION 2

**Proof.** [Proof of Lemma 2.1] Recall that for any one dimensional random variable $X$, we have

$$\int_0^\infty P(X \geq t)dt = \int_0^\infty \mathbb{E}[\mathbb{1}_{X \geq t}]dt = \mathbb{E}[\int_0^\infty \mathbb{1}_{X \geq t}dt] = \mathbb{E}[\int_0^X dt] = \mathbb{E}[X].$$

1. **First claim.**
   Assume $P(\|Y\| \geq R) \geq \frac{C}{R^p}$. Hence we know (where $s := t^{1/p}$, so that $dt = ps^{p-1}ds$)

$$\mathbb{E}[\|Y\|^p] = \int_0^\infty P(\|Y\|^p \geq t)dt = \int_0^\infty P(\|Y\| \geq t^{1/p})dt = \int_0^\infty P(\|Y\| \geq s)ps^{p-1}ds$$
$$\geq \int_0^\infty \frac{C}{s^p}ps^{p-1}ds = \int_0^\infty Cps^{-1}ds.$$

   The integral does not converge.

2. **Second claim.**
   Assume $P(\|Y\| \geq R) \leq \frac{C}{R^\beta}$.

$$\mathbb{E}[\|Y\|^p] = \int_0^\infty P(\|Y\|^p \geq t)dt = \int_0^\infty P(\|Y\| \geq t^{1/p})dt = \int_0^\infty P(\|Y\| \geq s)ps^{p-1}ds$$
$$\leq \int_0^\infty \frac{C}{s^\beta}ps^{p-1}ds = \int_0^\infty Cps^{p-1-\beta}ds.$$

   The integral converges iff $p - 1 - \beta < -1$, i.e., $\beta > p$.

$\square$

**Example 5.** Let $\mathcal{K} \subseteq \mathbb{R}^2$ be a triangle defined by the following inequalities:

$$100x_1 + 0.01x_2 \leq 1,$$
$$100x_1 - 0.01x_2 \leq 1,$$
$$-x_1 \leq 0.$$

Recall that for each constraint $a_i^T x \leq b_i$ we can define $\psi_i(x) = -\log(b_i - a_i^T x)$. Then the log-barrier is $\psi(x) = \sum_i \psi_i(x)$. Take derivative, we obtain $\nabla\psi(x) = \sum_i \frac{1}{b_i - a_i^T x}a_i$.

$$\nabla\psi(k_1, k_2) = \sum_i \frac{1}{b_i - a_i^T x}a_i = \frac{1}{1 - 100k_1 - 0.01k_2}\begin{bmatrix}100 \\ 0.01\end{bmatrix} + \frac{1}{1 - 100k_1 + 0.01k_2}\begin{bmatrix}100 \\ -0.01\end{bmatrix} + \frac{1}{k_1}\begin{bmatrix}-1 \\ 0\end{bmatrix}$$
$$= \begin{bmatrix}100(\frac{2-200k_1}{(1-100k_1-0.01k_2)(1-100k_1+0.01k_2)}) - \frac{1}{k_1} \\ 0.01(\frac{0.02k_2}{(1-100k_1-0.01k_2)(1-100k_1+0.01k_2)})\end{bmatrix}.$$

Consider two points $(k_1, k_2), (k_1, -k_2) \in \mathbb{R}^2$ in the dual space:

$$\|\nabla\psi(k_1, k_2) - \nabla\psi(k_1, -k_2)\| = \left\|\begin{bmatrix}0 \\ 0.01(\frac{0.04k_2}{(1-100k_1-0.01k_2)(1-100k_1+0.01k_2)})\end{bmatrix}\right\|.$$

Hence

$$\frac{\|\nabla\psi(k_1, k_2) - \nabla\psi(k_1, -k_2)\|}{\|(k_1, k_2)^T - (k_1, -k_2)^T\|}$$
$$= \frac{0.01(\frac{0.04k_2}{(1-100k_1-0.01k_2)(1-100k_1+0.01k_2)})}{2k_2} = 2 \times 10^{-4}\frac{1}{(1 - 100k_1 - 0.01k_2)(1 - 100k_1 + 0.01k_2)}.$$

When $(k_1, k_2) \to 0$, we have $\frac{\|\nabla\psi(k_1,k_2)-\nabla\psi(k_1,-k_2)\|}{\|(k_1,k_2)^T-(k_1,-k_2)^T\|} \to 2 \times 10^{-4}$.

The above example shows that, there are cases when the polytope is "ill-shaped", and leading to a very large $L_\psi$.

**Proof.** [Proof of Proposition 2.2] We have

$$\nabla\Psi(x) = \sum_{i=1}^m (-\phi_i(x))^{-\kappa}\nabla\phi_i(x) + x,$$
$$\nabla^2\Psi(x) = \kappa\sum_{i=1}^m (-\phi_i(x))^{-\kappa-1}\nabla\phi_i(x)\nabla\phi_i(x)^T + \sum_{i=1}^m (-\phi_i(x))^{-\kappa}\nabla^2\phi_i(x) + I.$$

Note that $\nabla^2 \phi_i(x) \succeq 0$ due to convexity of $\phi_i$. So we have $\nabla^2 \Psi(x) \succeq I$. It follows that
$$W_2(\nu, \mu) \leq W_{2,\Psi}(\nu, \mu).$$

Furthermore, $\nabla \Psi(x) = \sum_{i=1}^m (-\phi_i(x))^{-\kappa} \nabla \phi_i(x) + x$ so we know
$$\|\nabla \Psi(x)\| \leq \|x\| + \sum_{i=1}^m \|(-\phi_i(x))^{-\kappa} \nabla \phi_i(x)\| \leq \|x\| + \sum_{i=1}^m \frac{1}{\delta^\kappa} \|\nabla \phi_i(x)\|.$$

Since we assumed $\phi_i(x)$ are of bounded gradient, we know $\|\nabla \Psi(x)\| = \frac{C'}{\delta^\kappa}$ for some $C'$.

Denote
$$R_{\delta, \kappa} = \frac{C'}{\delta^\kappa} \geq \sup_{x \in \mathcal{K}_\delta} \|\nabla \Psi(x)\|.$$

Hence we know, $R_{\delta, \kappa}$ is such that $\{x \in \mathbb{R}^d : \|x\| \leq R_{\delta, \kappa}\} \supseteq \nabla \Psi(\mathcal{K}_\delta)$. It follows that
$$P(\|\nabla \Psi(X)\| \geq R_{\delta, \kappa}) \leq P(\mathcal{K} \backslash \mathcal{K}_\delta) \leq C_\mathcal{K} \delta^\beta = \frac{C}{R_{\delta, \kappa}^{\beta/\kappa}}.$$

where note that $\frac{C}{R_{\delta, \kappa}^{\beta/\kappa}} = \frac{C}{(\frac{C'}{\delta^\kappa})^{\beta/\kappa}} = C_\mathcal{K} \delta^\beta$. $\qquad \square$

# F    PROOFS FOR SECTION 3

We first provide some definitions related to conditional expectation in an abstract vector space. We follow the notation in Hytönen et al. (2016). Let $(S, \mathscr{A})$ be a measurable space, and $X$ a Banach space. $L^p(S; X)$ denote the linear space of all $\mu$-measurable functions from $S$ to $X$, with $\int_S \|f\|^p d\mu < \infty$. When $\mathscr{F}$ is a sub-$\sigma$-algebra of $\mathscr{A}$, $L^p(S; \mathscr{F}; X)$ represent the $L_p$ space w.r.t. $(S, \mathscr{F}, \mu|_\mathscr{F})$.

**Definition F.1.** *(Hytönen et al., 2016, Theorem 2.6.23 and Proposition 2.6.31)*

*If $\mu$ is $\sigma$-finite on the sub-algebra $\mathscr{F}$, then every $f \in L^1(S; X)$ admits a unique conditional expectation with respect to $\mathscr{F}$. It satisfies*
$$\int_F \mathbb{E}[f|\mathscr{F}] d\mu = \int_F f d\mu, \forall F \in \mathscr{F}.$$

*Furthermore, let $g \in L^0(S; \mathscr{F}; X_1)$, and that $f \in L^1(S; X_2)$ be $\sigma$-integrable over $\mathscr{F}$. Let $\beta : X_1 \times X_2 \to Y$ be a bounded bi-linear map. Then $\beta(g, f) \in L^0(S; Y)$ is $\sigma$-integrable over $\mathscr{F}$, and we have*
$$\mathbb{E}[\beta(g, f)|\mathscr{F}] = \beta(g, \mathbb{E}[f|\mathscr{F}]) \quad a.s.$$

**Proof.** [Proof of Proposition 3.1] In primal space, the corresponding interpolation would be
$$\frac{d}{dt} X_t = \frac{d}{dt} \nabla \psi^*(Z_t) = \nabla^2 \psi^*(Z_t) \frac{d}{dt} Z_t.$$

Recall that the two minimization problems are:
$$\min_v \mathbb{E}\left[\|v^P(X_t, t) - \frac{d}{dt} X_t\|_{g^P}^2\right] \quad \text{and} \quad \min_v \mathbb{E}\left[\|v^D(Z_t, t) - \frac{d}{dt} Z_t\|_{g^D}^2\right]$$
respectively.

Recall that $\nabla^2 \psi$ evaluated at $x$ is the inverse of $\nabla^2 \psi^*$ evaluated at $z = \nabla \psi(x)$, i.e., $\nabla^2 \psi(x) = (\nabla^2 \psi^*(\nabla \psi(x)))^{-1}$. Hence we obtain $\nabla^2 \psi(x) \circ \nabla^2 \psi^*(z) \frac{d}{dt} Z_t = \frac{d}{dt} Z_t$. Condition on $X_t = x$, we have
$$\|v^P(X_t, t) - \frac{d}{dt} X_t\|_{g^P}^2 = g^P\left(v^P(X_t, t) - \frac{d}{dt} X_t, v^P(X_t, t) - \frac{d}{dt} X_t\right)$$
$$= (\nabla^2 \psi(x))^2 \left(v^P(X_t, t) - \nabla^2 \psi^*(Z_t) \frac{d}{dt} Z_t, v^P(X_t, t) - \nabla^2 \psi^*(Z_t) \frac{d}{dt} Z_t\right)$$
$$= g^D\left(\nabla^2 \psi(x) v^P(x, t) - \frac{d}{dt} Z_t, \nabla^2 \psi(x) v^P(x, t) - \frac{d}{dt} Z_t\right) = \|\nabla^2 \psi(x) v^P(x, t) - \frac{d}{dt} Z_t\|_{g^D}^2.$$

Hence we get $\|v^P(x,t) - \frac{d}{dt}X_t\|^2_{g^P} = \|\nabla^2\psi(x)v^P(x,t) - \frac{d}{dt}Z_t\|^2_{g^D}$ or equivalently $\|v^D(z,t) - \frac{d}{dt}Z_t\|^2_{g^D} = \|\nabla^2\psi^*(z)v^D(z,t) - \frac{d}{dt}X_t\|^2_{g^P}$. So we get

$$v^D(z,t) = \nabla^2\psi(x)v^P(x,t), \quad v^P(x,t) = \nabla^2\psi^*(z)v^D(z,t).$$

The equivalence follows from the change of variable formula.

Now we show the last claim. Now consider $\mathcal{G}$ to be the sigma algebra corresponding to $X_t = x$. Note that each tangent space $T_x M$ is a Hilbert space, with Riemannian metric $g$. Then for any $Y$ (that is measurable in $\mathcal{G}$), we have

$$
\begin{aligned}
\mathbb{E}[\|\tfrac{d}{dt}X_t - Y\|^2_{g(x)}] &= \mathbb{E}[\|\tfrac{d}{dt}X_t - \mathbb{E}[\tfrac{d}{dt}X_t|X_t = x] + \mathbb{E}[\tfrac{d}{dt}X_t|X_t = x] - Y\|^2_{g(x)}] \\
&= \mathbb{E}[\|\tfrac{d}{dt}X_t - \mathbb{E}[\tfrac{d}{dt}X_t|X_t = x]\|^2_{g(x)}] + \mathbb{E}[\|\mathbb{E}[\tfrac{d}{dt}X_t|X_t = x] - Y\|^2_{g(x)}] \\
&\quad + 2\mathbb{E}[\langle \tfrac{d}{dt}X_t - \mathbb{E}[\tfrac{d}{dt}X_t|X_t = x], \mathbb{E}[\tfrac{d}{dt}X_t|X_t = x] - Y\rangle].
\end{aligned}
$$

Since $f := \mathbb{E}[\frac{d}{dt}X_t|X_t = x] - Y$ is measurable in $\mathcal{G}$, we have

$$
\begin{aligned}
\mathbb{E}[\langle \tfrac{d}{dt}X_t - \mathbb{E}[\tfrac{d}{dt}X_t|X_t = x], f\rangle] &= \mathbb{E}[\langle \tfrac{d}{dt}X_t, f\rangle] - \mathbb{E}[\langle\mathbb{E}[\tfrac{d}{dt}X_t|X_t = x], f\rangle] \\
&= \mathbb{E}[\langle \tfrac{d}{dt}X_t, f\rangle] - \langle\mathbb{E}[\mathbb{E}[\tfrac{d}{dt}X_t|X_t = x]], f\rangle = 0.
\end{aligned}
$$

where the last equality is by tower property (Hytönen et al., 2016, Proposition 2.6.33).

Hence we get

$$
\begin{aligned}
\mathbb{E}[\|\tfrac{d}{dt}X_t - Y\|^2_{g(x)}] &= \mathbb{E}[\|\tfrac{d}{dt}X_t - \mathbb{E}[\tfrac{d}{dt}X_t|X_t = x]\|^2_{g(x)}] + \mathbb{E}[\|\mathbb{E}[\tfrac{d}{dt}X_t|X_t = x] - Y\|^2_{g(x)}] \\
&\geq \mathbb{E}[\|\tfrac{d}{dt}X_t - \mathbb{E}[\tfrac{d}{dt}X_t|X_t = x]\|^2_{g(x)}], \forall Y \in \mathcal{G}.
\end{aligned}
$$

It follows that among all $Y$ being measurable in $\mathcal{G}$, the choice $Y = \mathbb{E}[\frac{d}{dt}X_t|X_t = x]$ minimize the problem. Hence $v^P(x,t) = \mathbb{E}[\frac{d}{dt}X_t|X_t = x]$. $\qquad\square$

# G  PROOFS FOR SECTION 4

We start with several intermediate results. Define $p_{t,x}(z_1) = \frac{(1+\frac{1}{\nu}\|\frac{x-tz_1}{1-t}\|^2)^{-\frac{\nu+d}{2}}p(z_1)}{\int_{\mathbb{R}^d}(1+\frac{1}{\nu}\|\frac{x-tz}{1-t}\|^2)^{-\frac{\nu+d}{2}}p(z)dz}$. Throughout this section, to make the notation compatible with $p_{t,x}(z_1)$, we use $p$ to denote the probability density function of the data distribution, supported on Euclidean space.

**Proposition G.1.** *Under Assumption 3 with $\alpha \geq 2d + \nu + 2$, there exists a constant $B$ that doesn' depend on $t, x$ s.t. for all $t \in [0, T]$,*

$$\mathbb{E}_{p_{t,x}(z_1)}[\|z_1\|^2] \leq \frac{B}{(1-T)^{\nu+d}}.$$

*In other words, we have that, for all $T \in (0, 1)$, there exists $B_1, B_2$ independent of $x$, so that*

$$\sup_{t\in[0,T]} \mathbb{E}_{p_{t,x}}[\|z_1\|] \leq B_1, \forall x,$$

$$\sup_{t\in[0,T]} \mathbb{E}_{p_{t,x}}[\|z_1\|^2] \leq B_2, \forall x.$$

**Proof.** [Proof of Proposition G.1]

To derive the desired upper bound, we aim to upper bound $p_{t,x}(z_1)$. We first derive a lower bound on the normalizing constant:

$$\int_{\mathbb{R}^d}(1+\frac{1}{\nu}\|\frac{x-tz_1}{1-t}\|^2)^{-\frac{\nu+d}{2}}p(z_1)dz_1$$

$$=\int_{\mathbb{R}^d}(\frac{(\frac{1-t}{t})^2}{(\frac{1-t}{t})^2+\frac{1}{\nu}\|\frac{x}{t}-z_1\|^2})^{\frac{\nu+d}{2}}p(z_1)dz_1 \geq \int_{\|z_1\|\leq R_0}(\frac{(\frac{1-t}{t})^2}{(\frac{1-t}{t})^2+\frac{1}{\nu}\|\frac{x}{t}-z_1\|^2})^{\frac{\nu+d}{2}}p(z_1)dz_1$$

$$\geq(\frac{(\frac{1-t}{t})^2}{(\frac{1-t}{t})^2+\frac{1}{\nu}(\frac{\|x\|}{t}+R_0)^2})^{\frac{\nu+d}{2}}(1-\frac{C'}{R_0^\beta}) \geq (\frac{(\frac{1-t}{t})^2}{(\frac{1-t}{t})^2+\frac{1}{\nu}(\frac{\|x\|}{t}+R_0)^2})^{\frac{\nu+d}{2}}\frac{1}{2}$$

$$\geq\frac{1}{2}(\frac{(1-t)^2}{(1-t)^2+\frac{1}{\nu}(\|x\|+tR_0)^2})^{\frac{\nu+d}{2}}.$$

We will split $\mathbb{R}^d$ into different regions, and derive upper bounds of $p_{t,x}(z_1)$ for each of them.

1. Region 1 $\|\frac{x}{t}-z_1\| \leq \frac{\|x\|}{2t}+R_0$. We remark that it suffices to consider $\frac{\|x\|}{2t} \geq 2R_0$ so that $\|\frac{x}{t}\|-\|z_1\| \leq \|\frac{x}{t}-z_1\| \leq \frac{\|x\|}{2t}+R_0$, which implies $\|z_1\| \geq \frac{\|x\|}{2t}-R_0 \geq R_0$.

   Otherwise, if $\frac{\|x\|}{2t} < 2R_0$, $\|z_1\|-\|\frac{x}{t}\| \leq \|\frac{x}{t}-z_1\| \leq \frac{\|x\|}{2t}+R_0$, i.e., we have that $\|z_1\| \leq 7R_0$, so that $\int_{B_{\frac{x}{t}}(R=\frac{\|x\|}{2t}+R_0)}\|z_1\|^2 p_{t,x}(z_1)dz_1 \leq 49R_0^2$ which is of constant order.

   Now since $\frac{\|x\|}{2t} \geq 2R_0$, we alternatively have

   $$\int_{\mathbb{R}^d}(1+\frac{1}{\nu}\|\frac{x-tz_1}{1-t}\|^2)^{-\frac{\nu+d}{2}}p(z_1)dz_1$$

   $$\geq\frac{1}{2}(\frac{(1-t)^2}{(1-t)^2+\frac{1}{\nu}(\|x\|+tR_0)^2})^{\frac{\nu+d}{2}} \geq \frac{1}{2}(\frac{(1-t)^2}{(1-t)^2+\frac{2}{\nu}\|x\|^2})^{\frac{\nu+d}{2}}.$$

   Using

   $$(1+\frac{1}{\nu}\|\frac{x-tz_1}{1-t}\|^2)^{-\frac{\nu+d}{2}}p(z_1)$$

   $$=\frac{1}{(1+\frac{1}{\nu}\|\frac{x-tz_1}{1-t}\|^2)^{\frac{\nu+d}{2}}}p(z_1) \leq \frac{1}{(1+\frac{1}{\nu}\|\frac{x-tz_1}{1-t}\|^2)^{\frac{\nu+d}{2}}}(\frac{C^{\frac{2}{\nu+d}}}{\|z_1\|^{\frac{2\alpha}{\nu+d}}})^{\frac{\nu+d}{2}}$$

   $$=(\frac{1}{(1+\frac{1}{\nu}\|\frac{x-tz_1}{1-t}\|^2)}\frac{C^{\frac{2}{\nu+d}}}{\|z_1\|^{\frac{2\alpha}{\nu+d}}})^{\frac{\nu+d}{2}} = (\frac{(\frac{1-t}{t})^2}{((\frac{1-t}{t})^2+\frac{1}{\nu}\|\frac{x}{t}-z_1\|^2)}\frac{C^{\frac{2}{\nu+d}}}{\|z_1\|^{\frac{2\alpha}{\nu+d}}})^{\frac{\nu+d}{2}}$$

   $$\leq(\frac{C^{\frac{2}{\nu+d}}}{\|z_1\|^{\frac{2\alpha}{\nu+d}}})^{\frac{\nu+d}{2}},$$

   we obtain

   $$p_{t,x}(z_1)=\frac{(1+\frac{1}{\nu}\|\frac{x-tz_1}{1-t}\|^2)^{-\frac{\nu+d}{2}}p(z_1)}{\int_{\mathbb{R}^d}(1+\frac{1}{\nu}\|\frac{x-tz}{1-t}\|^2)^{-\frac{\nu+d}{2}}p(z)dz} \leq 2(\frac{C^{\frac{2}{\nu+d}}}{\|z_1\|^{\frac{2\alpha}{\nu+d}}})^{\frac{\nu+d}{2}}(\frac{(1-t)^2}{(1-t)^2+\frac{2}{\nu}\|x\|^2})^{-\frac{\nu+d}{2}}$$

   $$=2(\frac{(1-t)^2+\frac{2}{\nu}\|x\|^2}{(1-t)^2}\frac{C^{\frac{2}{\nu+d}}}{\|z_1\|^{\frac{2\alpha}{\nu+d}}})^{\frac{\nu+d}{2}}.$$

When $\|\frac{x}{t} - z_1\| \leq \frac{\|x\|}{2t} + R_0$,

$$\int_{B_{\frac{x}{t}}(R=\frac{\|x\|}{2t}+R_0)} \|z_1\|^2 p_{t,x}(z_1) dz_1$$

$$\leq 2\int_{B_{\frac{x}{t}}(R=\frac{\|x\|}{2t}+R_0)} \|z_1\|^2 \left(\frac{(1-t)^2 + \frac{2}{\nu}\|x\|^2}{(1-t)^2} \frac{C^{\frac{2}{\nu+d}}}{\|z_1\|^{\frac{2\alpha}{\nu+d}}}\right)^{\frac{\nu+d}{2}} dz_1$$

$$\leq 2\text{Vol}(B_{\frac{x}{t}}(R=\frac{\|x\|}{2t}+R_0)) \sup_{B_{\frac{x}{t}}(R=\frac{\|x\|}{2t}+R_0)} \|z_1\|^2 \left(\frac{(1-t)^2 + \frac{2}{\nu}\|x\|^2}{(1-t)^2} \frac{C^{\frac{2}{\nu+d}}}{\|z_1\|^{\frac{2\alpha}{\nu+d}}}\right)^{\frac{\nu+d}{2}}$$

$$\leq 2C_B(\frac{\|x\|}{t})^d \sup_{B_{\frac{x}{t}}(R=\frac{3\|x\|}{4t})} \left(\frac{\frac{3}{\nu}\|x\|^2}{(1-T)^2} \frac{C^{\frac{2}{\nu+d}}}{\|z_1\|^{\frac{2\alpha-4}{\nu+d}}}\right)^{\frac{\nu+d}{2}}$$

$$\leq 2C_B\|x\|^d \frac{1}{t^d} \left(\frac{\frac{3}{\nu}}{(1-T)^2\|x\|^{-2}} \frac{C^{\frac{2}{\nu+d}}}{\|\frac{x}{4t}\|^{\frac{2\alpha-4}{\nu+d}}}\right)^{\frac{\nu+d}{2}}$$

$$\leq 2C_B\|x\|^d \frac{1}{t^d} \left(\frac{\frac{3}{\nu}C^{\frac{2}{\nu+d}}(4t)^{\frac{2\alpha-4}{\nu+d}}}{(1-T)^2\|x\|^{\frac{2\alpha-4-2\nu-2d}{\nu+d}}}\right)^{\frac{\nu+d}{2}}$$

$$= \|x\|^{2d+\nu+2-\alpha} t^{\alpha-2-d} \frac{1}{(1-T)^{\nu+d}} \left(2C_B 4^{\alpha-2}C(\frac{3}{\nu})^{\frac{\nu+d}{2}}\right),$$

where observe that $\frac{2\alpha-4}{\nu+d} - 2 = \frac{2\alpha-4-2\nu-2d}{\nu+d}$ and

$$\|x\|^d \left(\frac{1}{\|x\|^{\frac{2\alpha-4-2\nu-2d}{\nu+d}}}\right)^{\frac{\nu+d}{2}} = \|x\|^d \|x\|^{-\frac{2\alpha-4-2\nu-2d}{\nu+d}\frac{\nu+d}{2}} = \|x\|^{d-(\alpha-2-\nu-d)} = \|x\|^{2d+\nu+2-\alpha}.$$

To control the second moment so that it doesn't explode with $\|x\|$, we need $\alpha \geq 2d + \nu + 2$.

2. Region 2 $\|\frac{x}{t} - z_1\| \geq \frac{1}{2t}\|x\| + R_0$ and $\|z_1\| \geq 1$

For this case, we can have a sharper upper bound on $p_{t,x}(z_1)$.

$$(1 + \frac{1}{\nu}\|\frac{x-tz_1}{1-t}\|^2)^{-\frac{\nu+d}{2}} p(z_1)$$

$$= \frac{1}{(1 + \frac{1}{\nu}\|\frac{x-tz_1}{1-t}\|^2)^{\frac{\nu+d}{2}}} p(z_1) \leq \frac{1}{(1 + \frac{1}{\nu}\|\frac{x-tz_1}{1-t}\|^2)^{\frac{\nu+d}{2}}} \left(\frac{C^{\frac{2}{\nu+d}}}{\|z_1\|^{\frac{2\alpha}{\nu+d}}}\right)^{\frac{\nu+d}{2}}$$

$$= \left(\frac{1}{(1 + \frac{1}{\nu}\|\frac{x-tz_1}{1-t}\|^2)} \frac{C^{\frac{2}{\nu+d}}}{\|z_1\|^{\frac{2\alpha}{\nu+d}}}\right)^{\frac{\nu+d}{2}} = \left(\frac{(\frac{1-t}{t})^2}{((\frac{1-t}{t})^2 + \frac{1}{\nu}\|\frac{x}{t} - z_1\|^2)} \frac{C^{\frac{2}{\nu+d}}}{\|z_1\|^{\frac{2\alpha}{\nu+d}}}\right)^{\frac{\nu+d}{2}}$$

$$\leq \left(\frac{(\frac{1-t}{t})^2}{((\frac{1-t}{t})^2 + \frac{1}{\nu}(\frac{1}{2t}\|x\| + R_0)^2)} \frac{C^{\frac{2}{\nu+d}}}{\|z_1\|^{\frac{2\alpha}{\nu+d}}}\right)^{\frac{\nu+d}{2}} = \left(\frac{(1-t)^2}{((1-t)^2 + \frac{1}{\nu}(\frac{1}{2}\|x\| + tR_0)^2)} \frac{C^{\frac{2}{\nu+d}}}{\|z_1\|^{\frac{2\alpha}{\nu+d}}}\right)^{\frac{\nu+d}{2}}.$$

Hence

$$p_{t,x}(z_1) = \frac{(1 + \frac{1}{\nu}\|\frac{x-tz_1}{1-t}\|^2)^{-\frac{\nu+d}{2}} p(z_1)}{\int_{\mathbb{R}^d} (1 + \frac{1}{\nu}\|\frac{x-tz}{1-t}\|^2)^{-\frac{\nu+d}{2}} p(z) dz}$$

$$\leq 2\left(\frac{(1-t)^2}{(1-t)^2 + \frac{1}{\nu}(\frac{1}{2}\|x\| + tR_0)^2} \frac{C^{\frac{2}{\nu+d}}}{\|z_1\|^{\frac{2\alpha}{\nu+d}}}\right)^{\frac{\nu+d}{2}} \left(\frac{(1-t)^2}{(1-t)^2 + \frac{1}{\nu}(\|x\| + tR_0)^2}\right)^{-\frac{\nu+d}{2}}$$

$$= 2\left(\frac{(1-t)^2 + \frac{1}{\nu}(\|x\| + tR_0)^2}{(1-t)^2 + \frac{1}{\nu}(\frac{1}{2}\|x\| + tR_0)^2} \frac{C^{\frac{2}{\nu+d}}}{\|z_1\|^{\frac{2\alpha}{\nu+d}}}\right)^{\frac{\nu+d}{2}}.$$

We see that for $\|\frac{x}{t} - z_1\| \geq \frac{1}{2t}\|x\| + R_0$, $p_{t,x}(z_1)$ has a polynomial tail bound that doesn't depend on $x, t$. Thus $\int_{\|\frac{x}{t} - z_1\| \geq \frac{1}{2t}\|x\|+R_0 \text{ and } \|z_1\| \geq 1} \|z_1\|^2 p_{t,x}(z_1) dz_1$ can be bounded by some constant that doesn't depend on $x, t$:

$$\int_{\|\frac{x}{t} - z_1\| \geq \frac{1}{2t}\|x\|+R_0 \text{ and } \|z_1\| \geq 1} \|z_1\|^2 p_{t,x}(z_1) dz_1 \leq C' \int_{\|z_1\| \geq 1} \frac{1}{\|z_1\|^{\alpha-2}} dz_1.$$

The convergence of the integral is equivalent to the convergence of $\int_1^\infty r^{d-\alpha+1}$. When $\alpha \geq 2d + \nu + 2$, it converges.

3. Region 3 $\|\frac{x}{t} - z_1\| \geq \frac{1}{2t}\|x\| + R_0$ and $\|z_1\| \leq 1$.

   We simply have $\int_{\|\frac{x}{t} - z_1\| \geq \frac{1}{2t}\|x\| + R_0 \text{ and } \|z_1\| \leq 1} \|z_1\|^2 p_{t,x}(z_1) dz_1 \leq 1$.

$\square$

With the above Proposition, we can prove Lemma G.2 and G.3, which are the key ingredients in proving the Lipschitzness of $v$.

**Lemma G.2.** *Under Assumption 3, we have*

$$\|\nabla_x \mathbb{E}[Z_1 | Z_t = x]\| \leq \frac{\nu + d}{\nu} \frac{2\sqrt{\nu}}{1 - t} \mathbb{E}_{p_t(z_1|x)}[\|z_1\|] \leq \frac{\nu + d}{\nu} \frac{2\sqrt{\nu}}{1 - T} B_1, \forall t \in [0, T].$$

**Proof.** [Proof of Lemma G.2]

$$\nabla_x \mathbb{E}[Z_1 | Z_t = x] = \nabla_x \frac{\int_{\mathbb{R}^d} z_1 (1 + \frac{1}{\nu}\|\frac{x - tz_1}{1 - t}\|^2)^{-\frac{\nu+d}{2}} p(z_1) dz_1}{\int_{\mathbb{R}^d}(1 + \frac{1}{\nu}\|\frac{x - tz}{1 - t}\|^2)^{-\frac{\nu+d}{2}} p(z) dz}$$

$$= \int_{\mathbb{R}^d} z_1 \nabla_x \frac{(1 + \frac{1}{\nu}\|\frac{x - tz_1}{1 - t}\|^2)^{-\frac{\nu+d}{2}} p(z_1)}{\int_{\mathbb{R}^d}(1 + \frac{1}{\nu}\|\frac{x - tz}{1 - t}\|^2)^{-\frac{\nu+d}{2}} p(z) dz} dz_1$$

$$= \int_{\mathbb{R}^d} z_1 \left( \nabla_x(1 + \frac{1}{\nu}\|\frac{x - tz_1}{1 - t}\|^2)^{-\frac{\nu+d}{2}} \right)^T \frac{p(z_1) \int_{\mathbb{R}^d}(1 + \frac{1}{\nu}\|\frac{x - tz}{1 - t}\|^2)^{-\frac{\nu+d}{2}} p(z) dz}{\left( \int_{\mathbb{R}^d}(1 + \frac{1}{\nu}\|\frac{x - tz}{1 - t}\|^2)^{-\frac{\nu+d}{2}} p(z) dz \right)^2} dz_1$$

$$- \int_{\mathbb{R}^d} z_1 \left( \nabla_x \int_{\mathbb{R}^d}(1 + \frac{1}{\nu}\|\frac{x - tz}{1 - t}\|^2)^{-\frac{\nu+d}{2}} p(z) dz \right)^T \frac{(1 + \frac{1}{\nu}\|\frac{x - tz_1}{1 - t}\|^2)^{-\frac{\nu+d}{2}} p(z_1)}{\left( \int_{\mathbb{R}^d}(1 + \frac{1}{\nu}\|\frac{x - tz}{1 - t}\|^2)^{-\frac{\nu+d}{2}} p(z) dz \right)^2} dz_1.$$

Observe that

$$\nabla_x(1 + \frac{1}{\nu}\|\frac{x - tz_1}{1 - t}\|^2)^{-\frac{\nu+d}{2}}$$

$$= -\frac{\nu + d}{2}(1 + \frac{1}{\nu}\|\frac{x - tz_1}{1 - t}\|^2)^{-\frac{\nu+d}{2} - 1}(\nabla_x \frac{1}{\nu}\|\frac{x - tz_1}{1 - t}\|^2)$$

$$= -\frac{\nu + d}{2}(1 + \frac{1}{\nu}\|\frac{x - tz_1}{1 - t}\|^2)^{-\frac{\nu+d}{2} - 1}\frac{1}{\nu}(2\frac{x - tz_1}{1 - t})\frac{1}{1 - t}$$

$$= -(1 + \frac{1}{\nu}\|\frac{x - tz_1}{1 - t}\|^2)^{-\frac{\nu+d}{2}}\frac{\nu + d}{\nu}\frac{1}{1 + \frac{1}{\nu}\|\frac{x - tz_1}{1 - t}\|^2}(\frac{x - tz_1}{1 - t})\frac{1}{1 - t}.$$

Hence

$$\nabla_x \mathbb{E}[Z_1 | Z_t = x] = -\int_{\mathbb{R}^d} z_1 \left( (1 + \frac{1}{\nu}\|\frac{x - tz_1}{1 - t}\|^2)^{-\frac{\nu+d}{2}}\frac{\nu + d}{\nu}\frac{1}{1 + \frac{1}{\nu}\|\frac{x - tz_1}{1 - t}\|^2}(\frac{x - tz_1}{1 - t})\frac{1}{1 - t} \right)^T$$

$$\frac{p(z_1) \int_{\mathbb{R}^d}(1 + \frac{1}{\nu}\|\frac{x - tz}{1 - t}\|^2)^{-\frac{\nu+d}{2}} p(z) dz}{\left( \int_{\mathbb{R}^d}(1 + \frac{1}{\nu}\|\frac{x - tz}{1 - t}\|^2)^{-\frac{\nu+d}{2}} p(z) dz \right)^2} dz_1$$

$$+ \int_{\mathbb{R}^d} z_1 \left( \int_{\mathbb{R}^d}(1 + \frac{1}{\nu}\|\frac{x - tz}{1 - t}\|^2)^{-\frac{\nu+d}{2}}\frac{\nu + d}{\nu}\frac{1}{1 + \frac{1}{\nu}\|\frac{x - tz}{1 - t}\|^2}(\frac{x - tz}{1 - t})\frac{1}{1 - t} p(z) dz \right)^T$$

$$\frac{(1 + \frac{1}{\nu}\|\frac{x - tz_1}{1 - t}\|^2)^{-\frac{\nu+d}{2}} p(z_1)}{\left( \int_{\mathbb{R}^d}(1 + \frac{1}{\nu}\|\frac{x - tz}{1 - t}\|^2)^{-\frac{\nu+d}{2}} p(z) dz \right)^2} dz_1.$$

Recall that we use the notation $p_{t,x}(z_1) = \frac{(1+\frac{1}{\nu}\|\frac{x-tz_1}{1-t}\|^2)^{-\frac{\nu+d}{2}}p(z_1)}{\int_{\mathbb{R}^d}(1+\frac{1}{\nu}\|\frac{x-tz}{1-t}\|^2)^{-\frac{\nu+d}{2}}p(z)dz}$.

$$\nabla_x \mathbb{E}[Z_1|Z_t = x] = -\int_{\mathbb{R}^d} z_1 \left(\frac{\nu+d}{\nu}\frac{1}{1+\frac{1}{\nu}\|\frac{x-tz_1}{1-t}\|^2}(\frac{x-tz_1}{1-t})\frac{1}{1-t}\right)^T p_{t,x}(z_1)dz_1$$

$$+ \int_{\mathbb{R}^d} z_1 p_{t,x}(z_1)dz_1\left(\int_{\mathbb{R}^d}\frac{\nu+d}{\nu}\frac{1}{1+\frac{1}{\nu}\|\frac{x-tz}{1-t}\|^2}(\frac{x-tz}{1-t})\frac{1}{1-t}p(z|x)dz\right)^T.$$

In general, we have $\mathbb{E}[XY^T] - \mathbb{E}[X]\mathbb{E}[Y]^T = \mathbb{E}[(X-\mathbb{E}[X])(Y-\mathbb{E}[Y])^T]$. Let $X = z_1$ and $Y = \frac{\nu+d}{\nu}\frac{1}{1+\frac{1}{\nu}\|\frac{x-tz_1}{1-t}\|^2}(\frac{x-tz_1}{1-t})\frac{1}{1-t}$. To bound $\nabla_x\mathbb{E}[Z_1|Z_t = x]$, we consider any unit vector $v$:

$$v^T\nabla_x\mathbb{E}[Z_1|Z_t = x]v = \mathbb{E}[v^T(X-\mathbb{E}[X])\cdot v^T(Y-\mathbb{E}[Y])] \le \mathbb{E}[\|X-\mathbb{E}[X]\|\cdot\|Y-\mathbb{E}[Y]\|]$$

$$\le \mathbb{E}[(\|X\| + \|\mathbb{E}[X]\|)(\|Y\| + \|\mathbb{E}[Y]\|)] \le \mathbb{E}[\|X\|\|Y\|] + 3\mathbb{E}[\|X\|]\mathbb{E}[\|Y\|].$$

We have

$$\|Y\| = \frac{\nu+d}{\nu(1-t)^2}\frac{\|x-tz_1\|}{1+\frac{1}{\nu}\|\frac{x-tz_1}{1-t}\|^2} = \frac{\nu+d}{\nu}\frac{\|x-tz_1\|}{(1-t)^2+\frac{1}{\nu}\|x-tz_1\|^2}.$$

At $(1-t)^2 = \frac{1}{\nu}\|x-tz_1\|^2$, $\|Y\|$ reach maximum,

$$\sup_{z_1}\|Y\| = \frac{\nu+d}{\nu}\frac{\sqrt{\nu}}{2(1-t)}.$$

Therefore

$$\|\nabla_x\mathbb{E}[Z_1|Z_t = x]\| \le \frac{\nu+d}{\nu}\frac{2\sqrt{\nu}}{1-t}\mathbb{E}_{p_t(z_1|x)}[\|z_1\|].$$

$\square$

**Lemma G.3.** *Under Assumption 3 with $\alpha \ge 2d+\nu+2$, we have*

$$\|\frac{\partial}{\partial t}\mathbb{E}[Z_1|Z_t = x]\| \le \frac{\nu+d}{\nu}\frac{3\sqrt{\nu}}{2(1-t)^2}\left(\mathbb{E}[\|z_1\|^2] + 3\mathbb{E}[\|z_1\|]^2\right) \le \frac{\nu+d}{\nu}\frac{3\sqrt{\nu}}{2(1-T)^2}\left(B_2 + 3B_1^2\right), \forall t \in [0,T].$$

**Proof.** [Proof of Lemma G.3]

$$\frac{\partial}{\partial t}\mathbb{E}[Z_1|Z_t = x] = \frac{\partial}{\partial t}\frac{\int_{\mathbb{R}^d} z_1(1+\frac{1}{\nu}\|\frac{x-tz_1}{1-t}\|^2)^{-\frac{\nu+d}{2}}p(z_1)dz_1}{\int_{\mathbb{R}^d}(1+\frac{1}{\nu}\|\frac{x-tz}{1-t}\|^2)^{-\frac{\nu+d}{2}}p(z)dz}$$

$$= \int_{\mathbb{R}^d} z_1\nabla_x\frac{(1+\frac{1}{\nu}\|\frac{x-tz_1}{1-t}\|^2)^{-\frac{\nu+d}{2}}p(z_1)}{\int_{\mathbb{R}^d}(1+\frac{1}{\nu}\|\frac{x-tz}{1-t}\|^2)^{-\frac{\nu+d}{2}}p(z)dz}dz_1$$

$$= \int_{\mathbb{R}^d} z_1\left(\frac{\partial}{\partial t}(1+\frac{1}{\nu}\|\frac{x-tz_1}{1-t}\|^2)^{-\frac{\nu+d}{2}}\right)\frac{p(z_1)\int_{\mathbb{R}^d}(1+\frac{1}{\nu}\|\frac{x-tz}{1-t}\|^2)^{-\frac{\nu+d}{2}}p(z)dz}{\left(\int_{\mathbb{R}^d}(1+\frac{1}{\nu}\|\frac{x-tz}{1-t}\|^2)^{-\frac{\nu+d}{2}}p(z)dz\right)^2}dz_1$$

$$- \int_{\mathbb{R}^d} z_1\left(\frac{\partial}{\partial t}\int_{\mathbb{R}^d}(1+\frac{1}{\nu}\|\frac{x-tz}{1-t}\|^2)^{-\frac{\nu+d}{2}}p(z)dz\right)\frac{(1+\frac{1}{\nu}\|\frac{x-tz_1}{1-t}\|^2)^{-\frac{\nu+d}{2}}p(z_1)}{\left(\int_{\mathbb{R}^d}(1+\frac{1}{\nu}\|\frac{x-tz}{1-t}\|^2)^{-\frac{\nu+d}{2}}p(z)dz\right)^2}dz_1.$$

Observe that

$$\frac{\partial}{\partial t}(1+\frac{1}{\nu}\|\frac{x-tz_1}{1-t}\|^2)^{-\frac{\nu+d}{2}}$$

$$= -\frac{\nu+d}{2}(1+\frac{1}{\nu}\|\frac{x-tz_1}{1-t}\|^2)^{-\frac{\nu+d}{2}-1}(\frac{1}{\nu}\frac{\partial}{\partial t}\|\frac{x-tz_1}{1-t}\|^2)$$

$$= -\frac{\nu+d}{2}(1+\frac{1}{\nu}\|\frac{x-tz_1}{1-t}\|^2)^{-\frac{\nu+d}{2}-1}\frac{1}{\nu}(2\frac{x-tz_1}{1-t})^T\frac{x-z_1}{(1-t)^2}$$

$$= -(1+\frac{1}{\nu}\|\frac{x-tz_1}{1-t}\|^2)^{-\frac{\nu+d}{2}}\frac{\nu+d}{\nu}\frac{1}{1+\frac{1}{\nu}\|\frac{x-tz_1}{1-t}\|^2}\frac{1}{(1-t)^3}(\|x\|^2 - z_1^T x(1+t) + t\|z_1\|^2).$$

Hence

$$\nabla_x \mathbb{E}[Z_1 | Z_t = x]$$

$$= \int_{\mathbb{R}^d} z_1 \left( (1 + \frac{1}{\nu} \| \frac{x - tz_1}{1 - t} \|^2)^{-\frac{\nu+d}{2}} \frac{\nu + d}{\nu} \frac{1}{1 + \frac{1}{\nu} \| \frac{x - tz_1}{1-t} \|^2} \frac{1}{(1-t)^3} (\|x\|^2 - z_1^T x(1+t) + t\|z_1\|^2) \right)$$

$$\frac{p(z_1) \int_{\mathbb{R}^d} (1 + \frac{1}{\nu} \| \frac{x - tz}{1-t} \|^2)^{-\frac{\nu+d}{2}} p(z) dz}{\left( \int_{\mathbb{R}^d} (1 + \frac{1}{\nu} \| \frac{x - tz}{1-t} \|^2)^{-\frac{\nu+d}{2}} p(z) dz \right)^2} dz_1$$

$$- \int_{\mathbb{R}^d} z_1 \left( \int_{\mathbb{R}^d} (1 + \frac{1}{\nu} \| \frac{x - tz_1}{1-t} \|^2)^{-\frac{\nu+d}{2}} \frac{\nu+d}{\nu} \frac{1}{1 + \frac{1}{\nu} \| \frac{x-tz_1}{1-t} \|^2} \frac{1}{(1-t)^3} (\|x\|^2 - z_1^T x(1+t) + t\|z_1\|^2) p(z) dz \right)$$

$$\frac{(1 + \frac{1}{\nu} \| \frac{x - tz_1}{1-t} \|^2)^{-\frac{\nu+d}{2}} p(z_1)}{\left( \int_{\mathbb{R}^d} (1 + \frac{1}{\nu} \| \frac{x - tz}{1-t} \|^2)^{-\frac{\nu+d}{2}} p(z) dz \right)^2} dz_1.$$

Define $p_{t,x}(z_1) = \frac{(1 + \frac{1}{\nu} \| \frac{x - tz_1}{1-t} \|^2)^{-\frac{\nu+d}{2}} p(z_1)}{\int_{\mathbb{R}^d} (1 + \frac{1}{\nu} \| \frac{x - tz}{1-t} \|^2)^{-\frac{\nu+d}{2}} p(z) dz}$.

$$\frac{\partial}{\partial t} \mathbb{E}[Z_1 | Z_t = x]$$

$$= \int_{\mathbb{R}^d} z_1 \left( \frac{\nu+d}{\nu} \frac{1}{1 + \frac{1}{\nu} \| \frac{x-tz_1}{1-t} \|^2} \frac{1}{(1-t)^3} (\|x\|^2 - z_1^T x(1+t) + t\|z_1\|^2) \right) p_{t,x}(z_1) dz_1$$

$$- \int_{\mathbb{R}^d} z_1 p_{t,x}(z_1) dz_1 \left( \int_{\mathbb{R}^d} \frac{\nu+d}{\nu} \frac{1}{1 + \frac{1}{\nu} \| \frac{x-tz}{1-t} \|^2} \frac{1}{(1-t)^3} (\|x\|^2 - z^T x(1+t) + t\|z\|^2) p(z|x) dz \right).$$

Define $X = z_1, Y = \frac{\nu+d}{\nu} \frac{1}{1 + \frac{1}{\nu} \| \frac{x-tz_1}{1-t} \|^2} \frac{1}{(1-t)^3} (\|x\|^2 - z_1^T x(1+t) + t\|z_1\|^2)$. Note that

$$\|Y\| = \| \frac{\nu+d}{\nu} \frac{1}{1 + \frac{1}{\nu} \| \frac{x-tz_1}{1-t} \|^2} \frac{1}{(1-t)^3} (x - tz_1)^T (x - z_1) \|$$

$$\leq \frac{\nu+d}{\nu(1-t)^3} \frac{\|x - tz_1\| \|x - z_1\|}{1 + \frac{1}{\nu} \| \frac{x-tz_1}{1-t} \|^2} = \frac{\nu+d}{\nu} \frac{1}{1-t} \frac{\|x - tz_1\| \|x - z_1\|}{(1-t)^2 + \frac{1}{\nu} \|x - tz_1\|^2} \leq \frac{\nu+d}{\nu} \|z_1\|.$$

where observe that if $\|z_1\| \leq \frac{1}{2} \|x\|$, we have $\|x - z_1\| \leq 2\|x - tz_1\|$. Then $\|Y\| \leq \frac{\nu+d}{\nu^2} \frac{1}{1-t}$. If $\|z_1\| \geq \frac{1}{2} \|x\|$,

$$\|Y\| \leq \frac{\nu+d}{\nu} \frac{\sqrt{\nu}}{2(1-t)} \frac{1}{1-t} \|x - z_1\| \leq \frac{\nu+d}{\nu} \frac{\sqrt{\nu}}{2} \frac{1}{(1-t)^2} (\|x\| + \|z_1\|) \leq \|z_1\| \frac{\nu+d}{\nu} \frac{3\sqrt{\nu}}{2} \frac{1}{(1-t)^2}.$$

Recall: $\mathbb{E}[XY] - \mathbb{E}[X]\mathbb{E}[Y] = \mathbb{E}[(X - \mathbb{E}[X])(Y - \mathbb{E}[Y])]$. Therefore

$$\| \frac{\partial}{\partial t} \mathbb{E}[Z_1 | Z_t = x] \| \leq \mathbb{E}[v^T (X - \mathbb{E}[X])(Y - \mathbb{E}[Y])] \leq \mathbb{E}[\|X - \mathbb{E}[X]\| \cdot \|Y - \mathbb{E}[Y]\|]$$

$$\leq \mathbb{E}[(\|X\| + \|\mathbb{E}[X]\|)(\|Y\| + \|\mathbb{E}[Y]\|)] \leq \mathbb{E}[\|X\|\|Y\|] + 3\mathbb{E}[\|X\|]\mathbb{E}[\|Y\|]$$

$$\leq \frac{\nu+d}{\nu} \frac{3\sqrt{\nu}}{2} \frac{1}{(1-t)^2} \left( \mathbb{E}[\|z_1\|^2] + 3\mathbb{E}[\|z_1\|]^2 \right).$$

$\square$

The following Lemma will be used when analyzing the discretization error.

**Lemma G.4.** *Under Assumption 3 with $\alpha \geq 2d + \nu + 2$ and Assumption 2, there exists $D_3$ that depends polynomially in $\frac{1}{1-T}, d, \nu$ and $B_1, B_2, \mathbb{E}[\|Z_1\|\|^2], \mathbb{E}[\|Z_0\|\|^2]$ s.t.*

$$\mathbb{E}[\|v(Z_t, t) - v(Z_{t_i}, t_i)\|^2] \leq h^2 D_3.$$

**Proof.** [Proof of Lemma G.4]

By chain rule,

$$\frac{d}{dt}v(Z_t, t) = \frac{\partial}{\partial t}v(Z_t, t) + \frac{\partial}{\partial x}v(Z_t, t) \circ \frac{\partial}{\partial t}Z_t,$$

and therefore (note that $\frac{\partial}{\partial t}Z_t = v(Z_t, t)$)

$$\|\frac{d}{dt}v(Z_t, t)\| \leq \|\frac{\partial}{\partial t}v(Z_t, t)\| + \|\frac{\partial}{\partial x}v(Z_t, t)\| \cdot \|v(Z_t, t)\|.$$

Recall

$$\|\frac{\partial}{\partial t}v(x, t)\| \leq \frac{1}{(1-T)^2}\|x\| + \frac{1}{(1-T)^2}B_1 + \frac{1}{1-T}\frac{\nu+d}{\nu}\frac{3\sqrt{\nu}}{2(1-T)^2}\left(B_2 + 3B_1^2\right), \forall t \in [0, T]$$

$$\|\nabla_x v(x, t)\| \leq \frac{1}{1-T} + \frac{\nu+d}{\nu}\frac{2\sqrt{\nu}}{(1-T)^2}B_1, \forall t \in [0, T].$$

and

$$\|v(x, t)\| = \| - \frac{1}{1-t}x + \frac{1}{1-t}\mathbb{E}[Z_1|Z_t = x]\| \leq \frac{1}{1-T}\left(\|x\| + \|\mathbb{E}[Z_1|Z_t = x]\|\right)$$

$$\leq \frac{1}{1-T}\left(\|x\| + \mathbb{E}[\|Z_1\||Z_t = x]\right) = \frac{1}{1-T}\left(\|x\| + B_1\right).$$

Hence we have

$$\|\frac{d}{dt}v(Z_t, t)\| \leq \frac{1}{(1-T)^2}\|Z_t\| + \frac{1}{(1-T)^2}B_1 + \frac{1}{1-T}\frac{\nu+d}{\nu}\frac{3\sqrt{\nu}}{2(1-T)^2}\left(B_2 + 3B_1^2\right)$$

$$+ \left(\frac{1}{1-T} + \frac{\nu+d}{\nu}\frac{2\sqrt{\nu}}{(1-T)^2}B_1\right) \cdot \frac{1}{1-T}\left(\|Z_t\| + B_1\right) \forall t \in [0, T].$$

It follows that there exists $D_1, D_2$ (that depends polynomially in $\frac{1}{1-T}, d, \nu, B_1, B_2$) s.t.

$$\|\frac{d}{dt}v(Z_t, t)\|^2 \leq D_1\|Z_t\|^2 + D_2, \forall t \in [0, T].$$

Recall that $\text{Law}(Z_t) = \text{Law}(tZ_1 + (1-t)Z_0)$. Hence

$$\mathbb{E}[\|Z_t\|^2] = \mathbb{E}[\|tZ_1 + (1-t)Z_0\|^2] = t^2\mathbb{E}[\|Z_1\||^2] + (1-t)^2\mathbb{E}[\|Z_0\|^2] + 2t(1-t)\mathbb{E}[Z_0^T Z_1]$$

$$\leq 2\mathbb{E}[\|Z_1\||^2] + 2\mathbb{E}[\|Z_0\|^2].$$

which implies there exists $D_3$ (that depends polynomially in $\frac{1}{1-T}, d, \nu, B_1, B_2, \mathbb{E}[\|Z_1\||^2], \mathbb{E}[\|Z_0\||^2]$) s.t.

$$\mathbb{E}[\|\frac{d}{dt}v(Z_t, t)\|^2] \leq D_3.$$

By Jensen's inequality,

$$\mathbb{E}[\|v(Z_t, t) - v(Z_{t_i}, t_i)\|^2] = \mathbb{E}[\|\int_{t_i}^t \left(\frac{d}{ds}v(Z_s, s)\right) ds\|^2] \leq (t - t_i)\mathbb{E}[\int_{t_i}^t \left\|\frac{d}{ds}v(Z_s, s)\right\|^2 ds]$$

$$\leq h^2\mathbb{E}[\|\frac{d}{dt}v(Z_t, t)\|^2] \leq h^2 D_3.$$

$\square$

## G.1 PROOF OF PROPOSITION 4.1

**Proof.** [Proof of Proposition 4.1] Using Lemma G.2,

$$\|\nabla_x v(z, t)\| = \| - \frac{1}{1-t} I + \frac{1}{1-t} \nabla_x \mathbb{E}[Z_1 | Z_t = x]\|$$

$$\leq \frac{1}{1-T} + \frac{\nu + d}{\nu} \frac{2\sqrt{\nu}}{(1-T)^2} B_1, \forall t \in [0, T].$$

Notice that

$$\frac{\partial}{\partial t} v(z, t) = \frac{\partial}{\partial t} (-\frac{1}{1-t} z + \frac{1}{1-t} \mathbb{E}[Z_1 | Z_t = z])$$

$$= -\frac{1}{(1-t)^2} z + \frac{1}{(1-t)^2} \mathbb{E}[Z_1 | Z_t = z] + \frac{1}{1-t} \frac{\partial}{\partial t} \mathbb{E}[Z_1 | Z_t = z].$$

Using Lemma G.3, we have

$$\|\frac{\partial}{\partial t} v(z, t)\| \leq \frac{1}{(1-T)^2} \|z\| + \frac{1}{(1-T)^2} B_1 + \frac{1}{1-T} \frac{\nu + d}{\nu} \frac{3\sqrt{\nu}}{2(1-T)^2} \left( B_2 + 3B_1^2 \right), \forall t \in [0, T].$$

$\square$

## G.2 PROOF OF THEOREM 3

**Proof.** [Proof of Theorem 3]

Define

$$dZ_t = v(Z_t, t) dt, Z_0 \sim \pi_0,$$

$$d\overline{Y}_t = \hat{v}(\overline{Y}_{t_i}, t_i) dt, \overline{Y}_0 = Z_0.$$

By direct computation,

$$\frac{d}{dt} \|Z_t - \overline{Y}_t\|^2 = 2\langle Z_t - \overline{Y}_t, \frac{d}{dt} Z_t - \frac{d}{dt} \overline{Y}_t \rangle = 2\langle Z_t - \overline{Y}_t, v(Z_t, t) - \hat{G}(\overline{Y}_{t_i}, t_i) \rangle$$

$$= 2\langle Z_t - \overline{Y}_t, v(Z_t, t) - v(Z_{t_i}, t_i) \rangle + 2\langle Z_t - \overline{Y}_t, v(Z_{t_i}, t_i) - v(\overline{Y}_{t_i}, t_i) \rangle$$

$$+ 2\langle Z_t - \overline{Y}_t, v(\overline{Y}_{t_i}, t_i) - \hat{v}(\overline{Y}_{t_i}, t_i) \rangle.$$

Using Young's inequality, we can bound the rest of the terms as follows.

1. We bound the first term. By Lemma G.4,

$$2\mathbb{E}[\langle Z_t - \overline{Y}_t, v(Z_t, t) - v(Z_{t_i}, t) \rangle]$$

$$\leq L_1 \mathbb{E}[\|Z_t - \overline{Y}_t\|^2] + \frac{1}{L_1} \mathbb{E}[\|v(Z_t, t) - v(Z_{t_i}, t)\|^2]$$

$$\leq L_1 \mathbb{E}[\|Z_t - \overline{Y}_t\|^2] + \frac{1}{L_1} h^2 D_3.$$

2. We bound the second term.

$$2\mathbb{E}[\langle Z_t - \overline{Y}_t, v(Z_{t_i}, t_i) - v(\overline{Y}_{t_i}, t_i) \rangle]$$

$$\leq L_1 \mathbb{E}[\|Z_t - \overline{Y}_t\|^2] + \frac{1}{L_1} \mathbb{E}[\|v(Z_{t_i}, t_i) - v(\overline{Y}_{t_i}, t_i)\|^2]$$

$$\leq L_1 \mathbb{E}[\|Z_t - \overline{Y}_t\|^2] + \frac{1}{L_1} L_1^2 \mathbb{E}[\|Z_{t_i} - \overline{Y}_{t_i}\|^2].$$

Here we used Proposition 4.1.

3. We bound the third term. Recall that we assumed $\mathbb{E}[\|v(x,t) - \hat{v}(x,t)\|^2] \leq \varepsilon^2$. Then

$$2\langle Z_t - \overline{Y}_t, v(\overline{Y}_{t_i}, t_i) - \hat{v}(\overline{Y}_{t_i}, t_i)\rangle$$

$$\leq L_1 \mathbb{E}[\|Z_t - \overline{Y}_t\|^2] + \frac{1}{L_1}\mathbb{E}[\|v(\overline{Y}_{t_i}, t_i) - \hat{v}(\overline{Y}_{t_i}, t_i)\|^2]$$

$$\leq L_1 \mathbb{E}[\|Z_t - \overline{Y}_t\|^2] + \frac{1}{L_1}\varepsilon^2.$$

Together,

$$\frac{d}{dt}\mathbb{E}[\|Z_t - \overline{Y}_t\|^2] \leq 3L_1 \mathbb{E}[\|Z_t - \overline{Y}_t\|^2] + \frac{1}{L_1}\left(h^2 D_3 + L_1^2 \mathbb{E}[\|Z_{t_i} - \overline{Y}_{t_i}\|^2] + \varepsilon^2\right).$$

Define

$$K = h^2 D_3 + L_1^2 \mathbb{E}[\|Z_{t_i} - \overline{Y}_{t_i}\|^2] + \varepsilon^2.$$

Then

$$\mathbb{E}[\|Z_{t_{i+1}} - \overline{Y}_{t_{i+1}}\|^2]$$

$$\leq e^{3L_1 h}\mathbb{E}[\|Z_{t_i} - \overline{Y}_{t_i}\|^2] + \frac{3}{L_1}\int_{t_i}^{t_{i+1}} e^{3L_1(t_{i+1}-t)}(K)\,dt$$

$$\leq e^{3L_1 h}\mathbb{E}[\|Z_{t_i} - \overline{Y}_{t_i}\|^2] + \frac{e^{3L_1 h} - 1}{L_1^2}K$$

$$= e^{3L_1 h}\mathbb{E}[\|Z_{t_i} - \overline{Y}_{t_i}\|^2] + \frac{e^{3L_1 h} - 1}{L_1^2}(h^2 D_3 + \varepsilon^2) + (e^{3L_1 h} - 1)\mathbb{E}[\|Z_{t_i} - \overline{Y}_{t_i}\|^2]$$

$$\leq (2e^{3L_1 h} - 1)\mathbb{E}[\|Z_{t_i} - \overline{Y}_{t_i}\|^2] + \frac{e^{3L_1 h} - 1}{L_1^2}(h^2 D_3 + \varepsilon^2).$$

For $A_{i+1} \leq (2e^{3L_1 h} - 1)A_i + \frac{e^{3L_1 h}-1}{L_1^2}B$ with $A_0 = 0$, we have

$$A_n = \sum_{i=0}^{n-1}(2e^{3L_1 h} - 1)^i \frac{2e^{3L_1 h} - 1}{L_1^2}B = \frac{1 - (2e^{3L_1 h} - 1)^n}{1 - (2e^{3L_1 h} - 1)}\frac{e^{3L_1 h} - 1}{L_1^2}B \leq \frac{(2e^{3L_1 h} - 1)^n - 1}{2L_1^2}B.$$

In general, for $x \in [0,1]$ we have $e^x \leq 1 + 2x$. Hence $2e^{3L_1 h} - 1 \leq e^{3L_1 h} + 6L_1 h \leq 1 + 12L_1 h$. And we get $(2e^{3L_1 h} - 1)^n \leq (1 + 12L_1 h)^n \leq (1 + \frac{12L_1}{n})^n \leq e^{12L_1}$.

Hence

$$\mathbb{E}[\|Z_T - \overline{Y}_T\|^2] \leq \frac{e^{12L_1}}{L_1^2}(h^2 D_3 + \varepsilon^2).$$

This implies

$$W_2(\pi_T^D, \hat{\pi}_T^D) \leq \frac{e^{6L_1}}{L_1}\sqrt{h^2 D_3 + \varepsilon^2}.$$

Consequently,

$$W_2(\pi_1^D, \hat{\pi}_T^D) \leq \frac{e^{6L_1}}{L_1}\sqrt{h^2 D_3 + \varepsilon^2} + (1 - T)\sqrt{2(\mathbb{E}[\|Z_1\|^2] + \mathbb{E}[\|Z_0\|^2])}.$$

$\square$

### G.3 PROOF OF THEOREM 4

**Proof.** [Proof of Lemma 4.2] Note that we have

$$\|v^P(x_1, t) - P_{x_2}^{x_1} v^P(x_2, t)\|_{g^P(x_1)} = \|\nabla^2\Psi(x_1)\left(\nabla^2\Psi^*(z_1)v^D(z_1, t) - P_{x_2}^{x_1}\nabla^2\Psi^*(z_1)v^D(z_2, t)\right)\|_{g^D(z_1)}$$

$$= \|v^D(z_1, t) - \nabla^2\Psi(x_1)P_{x_2}^{x_1}\nabla^2\Psi^*(z_1)v^D(z_2, t)\|_{g^D(z_1)}$$

$$= \|v^D(z_1, t) - \nabla^2\Psi(x_1)\nabla^2\Psi^*(z_1)P_{z_2}^{z_1}v^D(z_2, t)\|_{g^D(z_1)}$$

$$= \|v^D(z_1, t) - v^D(z_2, t)\|_{g^D(z_1)},$$

where $P_x^y$ denotes parallel transport from $x$ to $y$. This proves the result. $\qquad\square$

**Lemma G.5.** *Under Assumption 4, For $\kappa \leq \frac{\gamma}{2d+\nu+2}$, we can guarantee Assumption 3 holds with $\alpha \geq 2d + \nu + 2$.*

**Proof.** Using the change of variable formula, together with the fact that the determinant of a matrix equals to the product of all its eigenvalues, we know

$$d\pi_{Euc}^P(x) = \sqrt{\det \nabla^2 \Psi(x)}d\pi_{Hess}^P(x) \geq d\pi_{Hess}^P(x),$$

where $\pi_{Euc}^P, \pi_{Hess}^P$ denotes the probability density function of the target distribution in primal space, under the Euclidean metric and squared Hessian metric, respectively. Furthermore, the isometric mapping from primal space to dual space guarantees that

$$\pi_{Euc}^P(x) \geq \pi^D(z).$$

Notice that

$$\sup_{x \in \mathcal{K}_\delta} \|\nabla \Psi(x)\| \leq \frac{C'}{\delta^\kappa}.$$

Since we assumed $\sup_{x \in \mathcal{K} \setminus \mathcal{K}_\delta} \pi_{Euc}^P(x) \leq C_{pdf}\delta^\gamma$, we have

$$\pi^D(z) \leq \pi_{Euc}^P(x) \leq C_{pdf}\delta^\gamma, \forall z \geq \frac{C'}{\delta^\kappa}.$$

Using $\delta^\gamma = (\frac{1}{(\frac{1}{\delta^\kappa})})^{\gamma/\kappa}$, we conclude that there exists some $C > 0$ s.t.

$$\pi^D(z) \leq \frac{C}{\|z\|^{\gamma/\kappa}}, \forall z \geq 1.$$

To guarantee $\gamma/\kappa \geq 2d + \nu + 2$, we need $\kappa \leq \frac{\gamma}{2d+\nu+2}$. $\qquad\square$

**Proof.** [Proof of Theorem 4] Using Lemma G.5, we know Assumption 3 holds with $\alpha \geq 2d + \nu + 2$. The result follows from applying Proposition 2.2 and Theorem 3. $\qquad\square$

