# OpenReview forum: "Mirror Flow Matching with Heavy-Tailed Priors for Generative Modeling on Convex Domains"
_ICLR.cc/2026/Conference — ICLR 2026 Poster_

### Official Review · Reviewer_hXik · 2025-10-31

**Soundness:** 3
**Presentation:** 2
**Contribution:** 2
**Rating:** 4
**Confidence:** 5

**Summary:**

This work proposes mirror flow matching, which addresses previous limitations of mirror diffusion models (Liu et al., 2023a) and presents a comprehensive theoretical framework for this approach. The key contributions include: (1) a modified mirror map designed to guarantee Lipschitz conditions, and (2) the use of a Student's t-distribution heavy-tailed prior for training the flow matching model in the transformed dual space. The empirical study demonstrates improvements compared to baseline methods, such as reflect flow matching and gauge flow matching.

**Strengths:**

1. The theoretical analysis is comprehensive and solid, including spatial and temporal regularity results and final distribution error bounds under discretization. These theoretical contributions may be of independent interest beyond this work.

**Weaknesses:**

1. The mirror map is limited to simple sets with closed-form expressions. Despite the comprehensive theoretical analysis of this framework, its practical applicability remains limited due to these restrictions.
2. The additional benefits in the empirical study of the proposed approach compared to Gaussian priors are minor. Moreover, the experiments lack a comparison to the traditional mirror diffusion models (in Tables 1 and 2), which serve as the main baseline work discussed throughout the paper.
3. The expression of the inverse mirror map is unclear. The authors should provide a table with explicit forms of both the forward and inverse mirror maps for different convex sets to improve clarity and reproducibility.
4. It is unclear how the Lipschitz conditions in Equation (1) are satisfied given Proposition 2.2, which only provides a bound on the expectation of the Lipschitz constant, while Equation (1) requires pointwise Lipschitz conditions.

**Questions:**

1. In the original mirror diffusion model work, the authors consider specific linear constraints with orthonormality conditions, which admit a closed-form inverse. However, such requirements are not explicitly stated in this work. Can the authors specify the forward and inverse mirror maps used in this work and clarify what conditions are necessary for closed-form inverses?
2. What is the role of the heavy-tailed prior distribution in Proposition 4.1? How does it contribute to the theoretical guarantees or empirical performance?
3. Why does RFM not achieve a 100% feasibility rate in Table 1?
4. In Proposition 3.1, what is the prior distribution for the primal and dual spaces, respectively?


Overall, this work is theoretically solid. I will adjust my rating upon clarification of the applicability and details of the mirror map stated above.

---

> ### Author Response · Authors · 2025-11-18
> **Official comment by Authors**
>
> We thank the reviewer for their thoughtful questions.
>
> Response to weaknesses:
>
> The mirror map is limited to simple sets with closed-form expressions. Despite the comprehensive theoretical analysis of this framework, its practical applicability remains limited due to these restrictions.
>  - We remark that even in this case, where the sets are defined with closed-form expressions, the problem is not completely solved. Performing flow matching for general (non-convex) constraint set is an interesting problem, we leave it for future work.
>
> The additional benefits in the empirical study of the proposed approach compared to Gaussian priors are minor. Moreover, the experiments lack a comparison to the traditional mirror diffusion models (in Tables 1 and 2), which serve as the main baseline work discussed throughout the paper.
>  - Experiments suggests proposed approach is better than Gaussian prior. See our updated experiments (Section 5.1 and Appendix D, new results are marked as blue color). We also added comparison with MDM. For $L_{2}$ ball example, MDM paper provided closed form formula to compute the log barrier and its inverse mirror map. Empirically, this induced a heavy tail, and the gradient exploded while training the neural network. Even with gradient clipping, the MDM doesn't perform well.
>
> It is unclear how the Lipschitz conditions in Equation (1) are satisfied given Proposition 2.2, which only provides a bound on the expectation of the Lipschitz constant, while Equation (1) requires pointwise Lipschitz conditions.
>  - In Proposition 2.2, we showed that for the proposed mirror map, Equation (1) holds, without taking any expectation. Consequently, we obtain the inequality for Wasserstein distance. We didn't mention nor prove a bound for "expectation of the Lipschitz constant''. We would hence, appreciate if you could clarify the question, if our response doesn't answer it.
>
> Response to questions:
>
> The expression of the inverse mirror map is unclear. The authors should provide a table with explicit forms of both the forward and
> inverse mirror maps for different convex sets to improve clarity and reproducibility.
> In the original mirror diffusion model work, the authors consider specific linear constraints with orthonormality conditions, which admit a closed-form inverse. However, such requirements are not explicitly stated in this work. Can the authors specify the forward and inverse mirror maps used in this work and clarify what conditions are necessary for closed-form inverses?
>  - Computing the inverse of mirror map is a convex optimization problem, we can solve it using existing approaches, e.g., Nesterov accelerated method. We observed that using Nesterov accelerated method, in most cases the output is inside of the feasible set, and for some extreme cases (rare), we can use python package cvxpy to handle them. We also remark that when the polytope doesn't have orthonormal constraints, the closed form expression from MDM paper doesn't apply.
>
> What is the role of the heavy-tailed prior distribution in Proposition 4.1? How does it contribute to the theoretical guarantees or empirical performance?
>
>  - First, note that theoretical analysis of error bounds for the sampling stage (of flow matching) requires the vector field $v$ to satisfy certain Lipschitz condition $x$ and $t$. In Proposition 4.1, we proved such Lipschitz condition.
>  - There are several existing works that theoretically justified such Lipschitz condition. They considered Gaussian prior, but to obtain the desired Lipschitz condition, they have to impose strong assumption on the target distribution. For example, Benton et al., 2024 and Zhou \& Liu, 2025 require the target distribution to be of bounded support, and Gao et al., 2024 require the traget distribution to be Gaussian-like. We mentioned this in our paper, see line 81-93.
>  - On the other hand, if we set the prior distribution to be Student t distribution, we can prove desired Lipschitz condition without such strong assumption on target distribution. This is exactly what Proposition 4.1 did.
>  - For some intuition, see Section 2.2 (Example 2 and the discussion for it) and Appendix C for an illustrative example.
>
> Why does RFM not achieve a 100% feasibility rate in Table 1?
>  - We adopted the code from Li et al. (2025). We realized that their code only considered one time reflection, while fully implementing RFM requires multiple reflection to guarantee feasibility. Thanks for pointing out this issue, and we fixed the bug. We updated the experimental result in the table. We remark that the computation time becomes much longer: we need to handle multiple time reflections.
>
> In Proposition 3.1, what is the prior distribution for the primal and dual spaces, respectively?
>  - Proposition 3.1 is not restricted to Gaussian or student t prior.
>
> **We greatly appreciate if could increase your scores if our response addressed your concerns; please let us know if you have further questions.**

---

> ### Author Response · Authors · 2025-11-27
>
> Dear Reviewer hXik
>
> As the deadline is approaching, we are eagerly waiting for your response to our answers.

---

### Official Review · Reviewer_9fkz · 2025-10-31

**Soundness:** 3
**Presentation:** 3
**Contribution:** 3
**Rating:** 6
**Confidence:** 3

**Summary:**

This work aims to improve the mirror flow matching algorithm — which is used to generate samples from a distribution supported on a convex domain — by proposing a new mirror map and using the Student-t distribution as the prior distribution.

They show on artificial data that the log-barrier transformation, which is commonly used as a mirror map, can induce heavy tails when pushing the target distribution into the dual space. To replace it, they introduce a new mirror map that ensures the existence of the moment of order p of the pushforward, by this new map, of probability measures satisfying the “boundary-measure condition.” This mirror map also has the property of being strongly convex, which allows transferring convergence guarantees from the dual space to the primal space.

They also illustrate, with a well-chosen 1D example, that for target distributions with heavy tails, the Gaussian prior is unsuited for the flow matching framework, and they propose using a Student-t prior instead. Still with the Student-t prior, they obtain Lipschitzness guarantees both in time and space for the true velocity field in the general flow matching framework when the target has a polynomial tail bound. This allows them to derive an upper bound on the Wasserstein distance between the true data distribution and the intermediate distributions generated by the learned velocity field of the general flow matching algorithm. Using the strong convexity property of the mirror map, they transfer this discretization bound from the dual space to the primal space.

They tested this new mirror flow matching approach on simulated data and on the FHQv2 dataset by generating watermarked images.

**Strengths:**

The paper is well written and a pleasure to read. The main new ideas concerning mirror flow matching — the mirror map and the new Student-t prior — are well motivated, and the arguments given to introduce each of them are fully valid.

In addition, the theoretical results on the Lipschitzness guarantees, both in time and in space, of the velocity field when the prior is a Student-t distribution (Proposition 4.1) are new and relevant for the general flow matching framework, as they require lighter constraints on the target distribution than prior works. The same holds for the discretization upper bound in Theorem 3, which is also relevant for the general flow matching framework.

**Weaknesses:**

While the theoretical sections are very rigorous, the experimental section is not and should be rewritten. The choice of baselines is not justified, and the baselines differ between Experiments 5.1 and 5.2. The choices of the hyperparameters $\kappa$ and $\nu$ are also not justified, and their values change between the two experiments. The architecture of the neural network used to estimate flow matching in the dual space is not mentioned. The comparison with other baselines is insufficiently detailed (e.g., the number of samples used to compute the MMD and KL divergence metrics in Table 1 is not specified). Finally, the notion of Feasibility (Table 1) is defined in the paper.

**Questions:**

- Proposition 2.2 assumes that the newly defined mirror map does not induce heavy tails and guarantees the existence of moments of the pushforward probabilities that initially (before being pushed forward by the mirror map) satisfy the boundary-measure condition. It would be interesting to see, in practice, for examples where this assumption is violated, whether the proposed mirror map induces lighter tails than the log-barrier map. You already verified this for the experiment described in Appendix A — do you observe the same result (i.e., that your proposed mirror map induces lighter tails than the log-barrier map) for the target distributions described in the experiments from Sections 5.1 and 5.2?

- Line 202: you claim, “This example highlights a key principle: when the target distribution is heavier-tailed than the prior, the conditional distribution is likely to have a mode that is dominant near $x_t$
 for some values of $t$” Do you know of other examples or references that could help generalize this principle beyond the pathological case presented in Example 2?

- Can you justify your choice of baselines for the experiments detailed in Sections 5.1 and 5.2?

- How did you set the hyperparameters $\kappa$ and $\nu$ in both experiments? What happens for other values of these hyperparameters? How sensitive is the algorithm to them?

- What is the architecture of the neural network used to parameterize the vector field trained in the dual space?

- Can you provide more information about the results reported in Table 1 (e.g., the number of samples used to compute the MMD and KL divergence metrics)? How is Feasibility defined?

- Finally, can you comment further on the fact that your results are not as good as those of the MDM baseline for the watermarked image generation task? Do you think this might be due to suboptimal choices of the hyperparameters $\kappa$ and $\nu$?

**Details Of Ethics Concerns:**

No concerns.

---

> ### Author Response · Authors · 2025-11-18
>
> We thank the reviewer for their thoughtful questions and positive evaluation.
>
> Response to weakness and questions:
>
> Proposition 2.2 assumes that the newly defined mirror map does not induce heavy tails and guarantees the existence of moments of the pushforward probabilities that initially (before being pushed forward by the mirror map) satisfy the boundary-measure condition.
>  - In Proposition 2.2, we showed that under boundary-measure condition, the newly defined mirror map provably doesn't induce heavy tails, and moments of the pushforward probabilities exists.
>
> It would be interesting to see, in practice, for examples where this assumption is violated, whether the proposed mirror map induces lighter tails than the log-barrier map. You already verified this for the experiment described in Appendix A — do you observe the same result (i.e., that your proposed mirror map induces lighter tails than the log-barrier map) for the target distributions described in the experiments from Sections 5.1 and 5.2?
>  - We added more figures in Appendix D, illustrating that for the polytope example, log barrier would induce heavy tails, but our proposed mirror map does not. The polytope constraint is the same as that of Section 5.1. We project the $10$ dimensional dual space distribution onto a two-dimensional subspace by selecting two coordinates in $\mathbb{R}^{10}$. The data distribution is a mixture of Gaussians, truncated to lie inside of the polytope. We observe that the log barrier would induce a significantly heavy tail. In contrast, for our mirror map, $\kappa = 0.7$ would induce a notably lighter tail. For $\kappa = 0.2$, the tail becomes almost invisible.
>
> Line 202: you claim, ``This example highlights a key principle: when the target distribution is heavier-tailed than the prior,
> the conditional distribution is likely to have a mode that is dominant near $x_{t}$ for some values of $t$". Do you know of other examples or references that could help generalize this principle beyond the pathological case presented in Example 2?
>  - The case presented in Example 2 is the motivation for us to explore theoretical properties of t-Flow. We showed in Proposition 4.1, that in general, such a principle holds. Using Student t distribution as prior, the vector field satisfy certain Lipschitz condition (in both $x$ and $t$ variable), provably.
>
> Can you justify your choice of baselines for the experiments detailed in Sections 5.1 and 5.2?
>  - As discussed in related works, for flow matching on constrained domains, classical method is reflected flow matching. We also added experiment for MDM in Section 5.1.
>
> How did you set the hyperparameters $\kappa$ and $\nu$ in both experiments? What happens for other values of these hyperparameters? How sensitive is the algorithm to them?
>  - We explored the effect of different parmeters $\nu, \kappa$ in Section 5.1, marked as blue. Empirically, we observed that a large $\nu$ would require a smaller $\kappa$, which is consistent with our theoretical findings.
>
> What is the architecture of the neural network used to parameterize the vector field trained in the dual space?
>  - We used a simple MLP network with 4 layers, and hidden layer size is $128$, with exponential linear unit (ELU) activation.
>
> Can you provide more information about the results reported in Table 1 (e.g., the number of samples used to compute the MMD and KL divergence metrics)? How is Feasibility defined?
>  - For each run, we generate $10,000$ samples. Feasibility is the percentage of generated samples that are in constraint set.
>
> Finally, can you comment further on the fact that your results are not as good as those of the MDM baseline for the watermarked image generation task? Do you think this might be due to suboptimal choices of the hyperparameters $\kappa$ and $\nu$?
>  - We improved our experimental results for section 5.2, see our updated results. When we initialize the model from a pretrained flow matching model, we obtain an FID (50k) of $3.14$ after $1.5$ hours of training. This value is similar to $3.05$ reported in the MDM paper. Furthermore, if we initialize both models from EDM (Karras et al., 2022) checkpoint, MDM achieves an FID (50k) value of $7.29$ after $13$ hours of training, whereas our approach achieves a significantly better FID (50k) value of $4.27$ after only $3$ hours of training (also achieves lower value of CMMD than MDM). See Table 3 for more details.

---

> > ### Comment · Reviewer_9fkz · 2025-11-25
> >
> > Dear authors,
> >
> > Sorry for the typo. In my first question, rather than writing “Proposition 2.2 assumes,” I meant to write “Proposition 2.2 ensures.”
> >
> > My main concern about this paper was the experimental section, which was not sufficiently detailed. The authors have done a great job revising this part by adding the missing information (neural network architectures, the number of samples used for Table 1, and the neural network training times). Another concern I had was the robustness of their method with respect to $\kappa$ and $\nu$. The authors have added an experiment (Figure 3) to show how performance varies with respect to these hyperparameters. They also managed to obtain better results on the watermarked images, which addressed my third major concern.
> >
> > Minor suggestions:
> >
> > I suggest that the authors write $\nu$ in the legend of Figure 3 rather than “df.”
> >
> > There is a small typo on line 461: “induced.”
> >
> > For these reasons, I have decided to increase my score.

---

> > > ### Author Response · Authors · 2025-11-27
> > >
> > > Dear reviewer 9fkz
> > >
> > > Thank you for your thoughtful review and positive evaluation. We will incorporate your suggestion in our revision.

---

### Official Review · Reviewer_H5Xj · 2025-11-01

**Soundness:** 4
**Presentation:** 4
**Contribution:** 3
**Rating:** 6
**Confidence:** 4

**Summary:**

The paper “Mirror Flow Matching with Heavy-Tailed Priors for Generative Modeling on Convex Domains”proposes a framework for generative modeling under convex constraints using flow matching enhanced by mirror maps and Student-t priors. The authors identify that traditional log-barrier mirror maps lead to heavy-tailed dual distributions and ill-posed dynamics, while Gaussian priors fail to align with such heavy-tailed targets. To address these issues, they introduce a regularized mirror map ensuring finite moments and strong convexity, coupled with Student-t priors that stabilize training and maintain well-behaved velocity fields. Theoretical contributions include proofs of spatial Lipschitzness, temporal regularity, and Wasserstein convergence bounds for flow matching with heavy-tailed priors, extended to the constrained (primal) domain. Empirically, the method achieves superior or competitive performance compared to reflection- and gauge-based baselines on constrained generative tasks—such as sampling within polytopes, L2  balls, and watermarked image generation—while guaranteeing feasibility and improving robustness on complex convex domains.

**Strengths:**

- The paper is rigorous and very well written.

- It makes relevant methodological contributions: particularly on how to perform flow matching with a mirror map framework, on how to choose the mirror map, and on how to set the initial distribution in order to handle target distributions with heavy tails, which is important as the distributions in the dual space can be heavy-tailed.

- The paper contains experiments in which the target distributions are over polytopes, and watermarked image generation experiments, although the images are low-quality.

**Weaknesses:**

- The applications of the work are not very compelling and may not justify the complexity of the framework.

- In the experimental section, the authors compare their performance to Mirror Diffusion Models, which is a previous work on using mirror maps for diffusion models. However, there is not comparison to Mirror Diffusion Models in the paper. Since Flow Matching and Diffusion Models are very related, it would be useful to clarify the differences between Mirror Flow Matching and Mirror Diffusion Models.

- More generally, the paper is lacking a related work section that places it in the context of previous works that leverage mirror maps for diffusion/flows.

- The paper can be considered incremental in that it is not the first diffusion/flows paper to rely on mirror maps, but it does clarify some design choices to make them work better.

**Questions:**

- How should the number of degrees of freedom of the Student t prior be set?

- What if the distribution has mass at the boundary of the constraint set? The mirror transformation would send such points to infinity, so how can we model them?

---

> ### Author Response · Authors · 2025-11-18
>
> We thank the reviewer for their thoughtful questions and positive evaluation.
>
> Response to weaknesses:
>
> The applications of the work are not very compelling and may not justify the complexity of the framework.
>  - Flow matching in constrainted domain is an important problem, and has received many attentions recently. For more details, see the discussion on Constrained Flow Matching and Related Works. Furthermore, the results in Section 4.1 holds for Euclidean flow matching. To the best of our knowledge, this result is the first to provide an convergence analysis for flow matching without strong assumptions. For more details, see "Theoretical Challenges".
>
> In the experimental section, the authors compare their performance to
> Mirror Diffusion Models, which is a previous work on using mirror maps for
> diffusion models. However, there is not comparison to Mirror Diffusion Models in the paper.
> Since Flow Matching and Diffusion Models are very related,
> it would be useful to clarify the differences between Mirror Flow Matching and
> Mirror Diffusion Models.
>  - Maybe we don't fully understand this question. For mirror diffusion model, it belongs to the class of diffusion model (MDM), and they used DDPM framework. For mirror flow matching, it belongs to the class of flow matching methods. The main difference, from a high level idea, is that our approach is a flow matching method, while MDM is DDPM. Also, we updated our experiments. The new results are marked as blue. In Section 5.1, we compared our approach against MDM, and evaluated the performance of our approach under different choices of parameters $\nu, \kappa$.
>
> More generally, the paper is lacking a related work section that places it in the context of previous works
> that leverage mirror maps for diffusion/flows.
>  - The discussion for related works was in "Constrained Flow Matching". We added a paragraph "Related works", marked as blue.
>
> Response to questions:
>
> How should the number of degrees of freedom of the Student t prior be set?
>  - Theoretically, we can take any $\nu$ as long as the second moment of the Student t distribution is finite. But note that theoretically, the larger $\nu$ being set, the smaller $\kappa$ we need. We also tested the effect of different $\nu$ empirically, the figure can be found in Section 5.1. The empirical result is consistent with our theoretical findings.
>
> What if the distribution has mass at the boundary of the constraint set? The mirror transformation would send such points to infinity, so how can we model them?
>  - We require the constraint set to be open. In Proposition 2.2, we wrote $\phi_{i}(x) < 0, \forall i$. We corrected the typo in Appendix B. For example, for polytope constraint, we need $Ax < b$.

---

> ### Author Response · Authors · 2025-11-27
>
> Dear Reviewer H5Xj
>
> As the deadline is approaching, we are eagerly waiting for your response to our answers.

---

### Author Response · Authors · 2025-11-30

We are submitting this response to summarize our contributions, rebuttals and the reviewers' feedback (so far).

**Main Contribution.**

 - We proposed a framework for generative modeling under convex constraints using flow matching enhanced by mirror maps and Student-t priors.

 - We introduced a regularized mirror map ensuring finite moments and strong convexity, coupled with Student-t priors that stabilize training and maintain well-behaved velocity fields.

 - We provided theoretical guarantees on spatial Lipschitzness, temporal regularity of the vector field when the prior is a Student-t distribution (Proposition 4.1). The theoretical contributions in Proposition 4.1 are new and relevant for the general **Euclidean** flow matching framework and may be of independent interest beyond this work, as they require lighter constraints on the target distribution than prior works.

 - We further provided Wasserstein convergence bounds for flow matching with heavy-tailed priors, extended to the constrained (primal) domain (Theorem 3 and Theorem 4).

 - The empirical study demonstrates improvements compared to baseline methods, such as reflect flow matching and gauge flow matching.

**Rebuttal.**

 - We made the discussion on related work explicit (addressing comment by reviewer H5Xj), starting from line 49.

 - We significantly expanded our experimental comparison with Mirror Diffusion Model in section 5.1, addressing the comments by reviewer hXik.

 - We added experiments exploring the effect of different parameters $\nu, \kappa$ in Section 5.1, Figure 3; we also improved our experimental result in Section 5.2 on real data experiment (addressing the comment by reviewer 9fkz).  Reviewer 9fkz replied to our rebuttals on Nov. 25 and acknowledged that we did a great job addressing their main concerns. Reviewer 9fkz increased their score from 6 to 8.

 - While reviewer hXik acknowledged our theoretical contribution is solid and may be of independent interest beyond this work, they asked about (question 2) the role of the heavy-tailed prior distribution in Proposition 4.1. We answered their question in detail, including 1) why Lipschitzness of the vector field is important; 2) prior works require stronger assumptions on the target distribution to prove Lipschitzness of the vector field, compared with our work; 3) the use of student-t prior allows us to prove Proposition 4.1 under lighter assumptions. We remark that this is actually one of our main contributions, acknowledged by reviewer 9fkz in the initial review.

 - Also, reviewer hXiK wrote "It is unclear how the Lipschitz conditions in Equation (1) are satisfied given Proposition 2.2..." (weakness 4). We answered this weakness by clarifying what Proposition 2.2 actually proved, as well as the fact that we didn't mention "expectation of the Lipschitz constant''.

---

### Meta-Review · Area_Chair_LfjK · 2026-01-08

**Summary:**

This paper introduces Mirror Flow Matching (MFM), a principled framework for generative modeling under convex constraints using mirror maps and Student-t priors. The authors identify instability issues caused by standard log-barrier mirror maps and Gaussian priors, which induce heavy-tailed dual distributions and ill-posed dynamics. To address this, they propose a regularized, strongly convex mirror map ensuring finite moments, together with heavy-tailed priors that yield well-behaved velocity fields. The paper provides theoretical guarantees on spatial and temporal regularity of the vector field and establishes Wasserstein convergence bounds for flow matching with heavy-tailed priors, which are then transferred back to the constrained primal domain. Experiments on constrained sampling tasks and watermarked image generation demonstrate improved feasibility and robustness compared to reflection-, gauge-, and mirror-diffusion-based baselines.

**Reviewer Concerns:**

The main reviewer concerns centered on insufficient experimental detail, missing comparisons to Mirror Diffusion Models, unclear feasibility metrics, and lack of hyperparameter sensitivity analysis. These were substantively addressed in the rebuttal through expanded experiments, added baselines, clarified implementation details, and new ablations. Multiple reviewers explicitly acknowledged these improvements and increased their scores.

The approach remains most directly applicable to convex domains with tractable mirror maps or numerical inversion, limiting immediate generality. Empirical gains over strong baselines are generally consistent but moderate, and the demonstrated applications remain relatively specialized. These limitations are acknowledged by the authors and do not detract from the correctness or clarity of the contribution.

**Reviewer Scores:**

At least one reviewer raised their scores (from 6 to 8), and others signaled satisfaction after the rebuttal. The overall trajectory across reviews is net positive, with no major unresolved objections.

---

### Decision · Program_Chairs · 2026-01-26

Accept (Poster)